# Lateral hypothalamic proenkephalin neurons drive threat-induced overeating associated with a negative emotional state

In-Jee You[1,2,4], Yeeun Bae[1,2,3,4], Alec R. Beck[1,2] & Sora Shin [1,2,3] ✉

Psychological stressors, like the nearby presence of a predator, can be strong enough to induce physiological/hormonal alterations, leading to appetite changes. However, little is known about how threats can alter feeding-related hypothalamic circuit functions. Here, we found that proenkephalin (Penk)-expressing lateral hypothalamic (LH[Penk]) neurons of mice exposed to predator scent stimulus (PSS) show sensitized responses to high-fat diet (HFD) eating, whereas silencing of the same neurons normalizes PSS-induced HFD over-consumption associated with a negative emotional state. Downregulation of endogenous enkephalin peptides in the LH is crucial for inhibiting the neuronal and behavioral changes developed after PSS exposure. Furthermore, elevated corticosterone after PSS contributes to enhance the reactivity of glucocorticoid receptor (GR)-containing LH[Penk] neurons to HFD, whereas pharmacological inhibition of GR in the LH suppresses PSS-induced maladaptive behavioral responses. We have thus identified the LH[Penk] neurons as a critical component in the threat-induced neuronal adaptation that leads to emotional overconsumption.

Emotional eating is defined as the excessive consumption of comfort foods in response to negative emotional valence instead of physical hunger[1,2]. For individuals exposed to life-threatening events in uncontrollable environments, emotional eating can provide relief from distress and tension, helping them to regain a sense of control[3]. Studies have reported that stress-induced maladaptive responses of feeding-related neural circuits or neuroendocrine systems may cause appetite changes[4–6], but it remains unclear how specific neural circuits or hormonal dynamics are involved and how their interactions adapt in response to threatening stimuli that trigger emotional eating.

The lateral hypothalamus (LH) comprises a large portion of the hypothalamus and contains several genetically distinct cell populations[7]. As expected from its considerable cellular heterogeneity, LH circuits have been implicated in various primitive behavior patterns, including feeding, emotion, predation/evasion in predator-prey relationships, and stress responses[8–13]. Previous studies suggest that hypocretin (Hcrt)-expressing LH neurons can modulate the intensity with which the hypothalamic-pituitary-adrenal (HPA) axis responds to stress, typically leading to fight-or-flight responses[11]. In addition, leptin receptor (Lepr)-expressing LH circuits are involved in the central regulation of context-specific palatable food over-consumption or appetitive behaviors toward non-drug reinforcers[5,14]. Despite this wealth of data, however, the key LH neuronal populations that respond to threat-triggered negative emotions and transduce those signals into strong preferences for high-calorie palatable foods remain poorly characterized.

Proenkephalin (Penk), a precursor of the endogenous opioid peptides met- and leu-enkephalin, is widely expressed in multiple brain regions, where it modulates neuropeptidergic systems that control various physiological and emotional processes[15–17]. For example, Penk-expressing neurons in the nucleus accumbens medial shell (mNAcsh) potentiate sucrose consumption by altering the presynaptic neuronal

[1]Fralin Biomedical Research Institute at VTC, Roanoke, VA, USA. [2]FBRI Center for Neurobiology Research, Roanoke, VA, USA. [3]Department of Human Nutrition, Foods, and Exercise, Virginia Polytechnic Institute and State University, Blacksburg, VA, USA. [4]These authors contributed equally: In-Jee You, Yeeun Bae. ✉e-mail: srshin@vtc.vt.edu

activity of μ-opioid peptide receptor (MOR)-expressing lateralized population of dorsal raphe nucleus[18]. In the paraventricular nucleus (PVN), the Penk-mediated opioid mechanism attenuates HPA axis activity and behavioral stress responses[19], while in the basolateral amygdala, Penk plays a role in eliciting chronic stress-induced anxiety[20]. The LH is also known to express Penk[21,22], but to date, the roles of the endogenous enkephalinergic systems in the LH circuitry in mediating post-threat effects like emotional eating have not been explored.

Stress hormones, particularly corticosterone (CORT), can exert appetite-stimulating effects[4,23,24]. Activation of the HPA axis elevates CORT levels following the onset of stressful events. This increase typically returns to baseline levels within hours or days after the threatening situation ends. Given that CORT receptors are highly expressed in multiple hypothalamic subregions[25], it is plausible that threat-triggered transient elevations of CORT interact with specific hypothalamic neurons to sensitize the neuronal responses to palatable foods. Recent studies have shown that CORT activates agouti-related peptide (AgRP)-expressing arcuate nucleus (Arc) neurons to drive homeostatic hunger[23]. However, beyond homeostatic regulation, it is still unknown how threat-induced increases in CORT levels produce long-term changes in LH neuronal properties that lead to emotionally triggered palatable food overconsumption.

Here, we have explored the role of LH^Penk neurons in emotional eating in mice exposed to naturally threatening predator-scent stimulus (PSS) with cat odor. We found that mice exhibited a negative emotional state and overconsumption of high-fat diet (HFD) 24 h after PSS exposure. Microendoscopic calcium imaging of LH^Penk neurons revealed that PSS exposure rendered LH^Penk neurons more sensitive to HFD, thereby displaying potentiated in vivo activity of LH^Penk neurons in response to HFD eating bouts. We also found that chemogenetic/optogenetic activation of LH^Penk neurons recapitulated both HFD overconsumption and the negative emotional state, whereas silencing of the same neurons abolished those behaviors following PSS. Disruption of enkephalin in the LH reversed the sensitized responses of LH^Penk neurons to HFD associated with emotional overeating in PSS mice. Moreover, MOR activity in the lateral periaqueductal gray (LPAG), one of the downstream structures of LH^Penk neurons, was necessary for promoting PSS-induced HFD overconsumption. We further report that PSS significantly elevated serum CORT levels and that LH^Penk neurons likely respond to PSS-induced CORT because they predominantly express glucocorticoid receptors (GR). Indeed, we found that pretreatment with CORT increased HFD intake and LH^Penk neuronal reactivity to HFD, mimicking the effects of PSS. Finally, we showed that pharmacological inhibition of GRs in the LH attenuated PSS-induced HFD overconsumption, possibly by suppressing negative emotional valence. Our results identified LH^Penk neurons as an important convergent target of the negative emotional valence and disordered eating behaviors that often follow life-threatening events.

## Results

### Mice exhibit HFD overconsumption at 24 h post-PSS exposure

To understand how threatening stimuli trigger emotional stress responses, we used a PSS paradigm[26,27] in which mice (8–10 weeks old) are exposed to a volatile predator cue (e.g., cat odor) that signifies imminent danger (Fig. 1a). To assess the acute impact of PSS on the development of negative emotions such as aversion, we exposed individual mice to a confined chamber with a PSS zone (center) and a non-PSS zone (periphery). We found that, in the presence of PSS, mice spend significantly more time in the periphery, maximizing their distance from the center with no accompanying changes in locomotion (Fig. 1b–d). This suggests that mice develop aversive emotional responses that prompt avoidance behaviors upon encountering PSS.

Maladaptive eating behaviors, including the emotional overeating of palatable foods, are often considered coping mechanisms for

individuals who have been primed with negative emotions after life-threatening events[3,28]. Given that the cellular or circuit-level changes underlying the stress-induced maladaptive behaviors require time to take hold[29,30], there is likely a delay between PSS exposure and the onset of emotional eating. To determine whether such a delay is necessary for mice to develop emotional overeating after PSS, we subjected mice to HFD (60% kcal% fat) either immediately or 24 h after PSS exposure (Fig. 1e). Immediately after PSS exposure, we did not observe any changes in the HFD consumption regardless of PSS history. In contrast, 24 h after PSS exposure, the PSS mice consumed much more HFD in 2.5 h than mice that were never exposed to PSS (Fig. 1f–i). We did not see any PSS-dependent differences in normal chow (NC) intake among various groups of mice (Fig. 1f, g). These data suggest PSS enhances preference for palatable food in mice, particularly 24 h after exposure.

Emotional eating is triggered by negative emotions that arise and persist after stressful events. We thus asked whether PSS induces negative emotional valence in addition to HFD overconsumption at 24 h post-exposure. Using a classical conditioning paradigm, we found that mice exhibited a conditioned place avoidance (CPA) of cues previously paired with PSS and that this result is independent of any changes in locomotor activity (Fig. 1j–l). This suggests mice tend to retain negative emotions at 24 h post-PSS exposure. Together, these data support the hypothesis that PSS induces a delayed behavioral response that comprises a tendency to overconsume HFD accompanied by negative emotional valence.

### Anatomical identification of LH^Penk neurons

Exposure to stressful situations can promote palatable food consumption via increasing the hypothalamic circuit activity involved in eating behaviors[5,6]. To determine whether PSS promotes emotional eating by increasing hypothalamic activity in response to HFD, we quantified c-fos expression in several hypothalamic subregions after PSS exposure. We found that, both in the LH and PVN, mice exposed to PSS a day before showed remarkably enhanced c-fos expression in response to HFD compared to non-PSS controls (Supplementary Fig. 1a–d). While the dorsomedial hypothalamus (DMH) showed a modest increase in c-fos induction under the same condition, we did not detect significant changes in other hypothalamic subregions, such as the ventromedial hypothalamus (VMH) or Arc (Supplementary Fig. 1e–j). With our discovery that PSS mice overconsume HFD (Fig. 1g), these c-fos data suggest that the LH or PVN could mediate PSS-induced emotional overeating.

Penk, an endogenous opioid polypeptide hormone that produces enkephalin, has been implicated in several emotional behaviors, including fear conditioning, anxiety, and responses to stress[17,31]. Using Penk-Cre mice crossed with the Ai14 reporter mouse line, we found that Penk is highly expressed in the hypothalamic subregions, including the LH, PVN, and Arc (Fig. 2a, b). Because the LH is a neuroanatomical hub responsible for regulating diverse primitive behavioral states, such as feeding, as well as approach and avoidance behaviors[7,12,13], we hypothesized that LH^Penk neuronal population is likely to be responsible for PSS-induced emotional overeating. In the LH, however, there are other cell-type of subpopulations that are already known to be involved in controlling behavioral stress responses and feeding behaviors[5,11,32], including pro-melanin concentrating hormone (Pmch)-, Hcrt-, and Lepr-expressing neurons. Therefore, we first sought to examine whether the LH^Penk neurons are anatomically distinct from those subpopulations. Using a dual fluorescent in situ hybridization (FISH) experiment, we found only minor overlaps between LH^Penk neurons and other cell-type of LH neurons expressing the Pmch, Hcrt, or Lepr, while the large proportions of LH^Penk neurons (76–90%) represent a non-overlapping and discrete population (Fig. 2c–h and Supplementary Fig. 2a–c). Furthermore, using FISH to probe mRNA expression of VGAT and vGlut2, which are GABAergic and

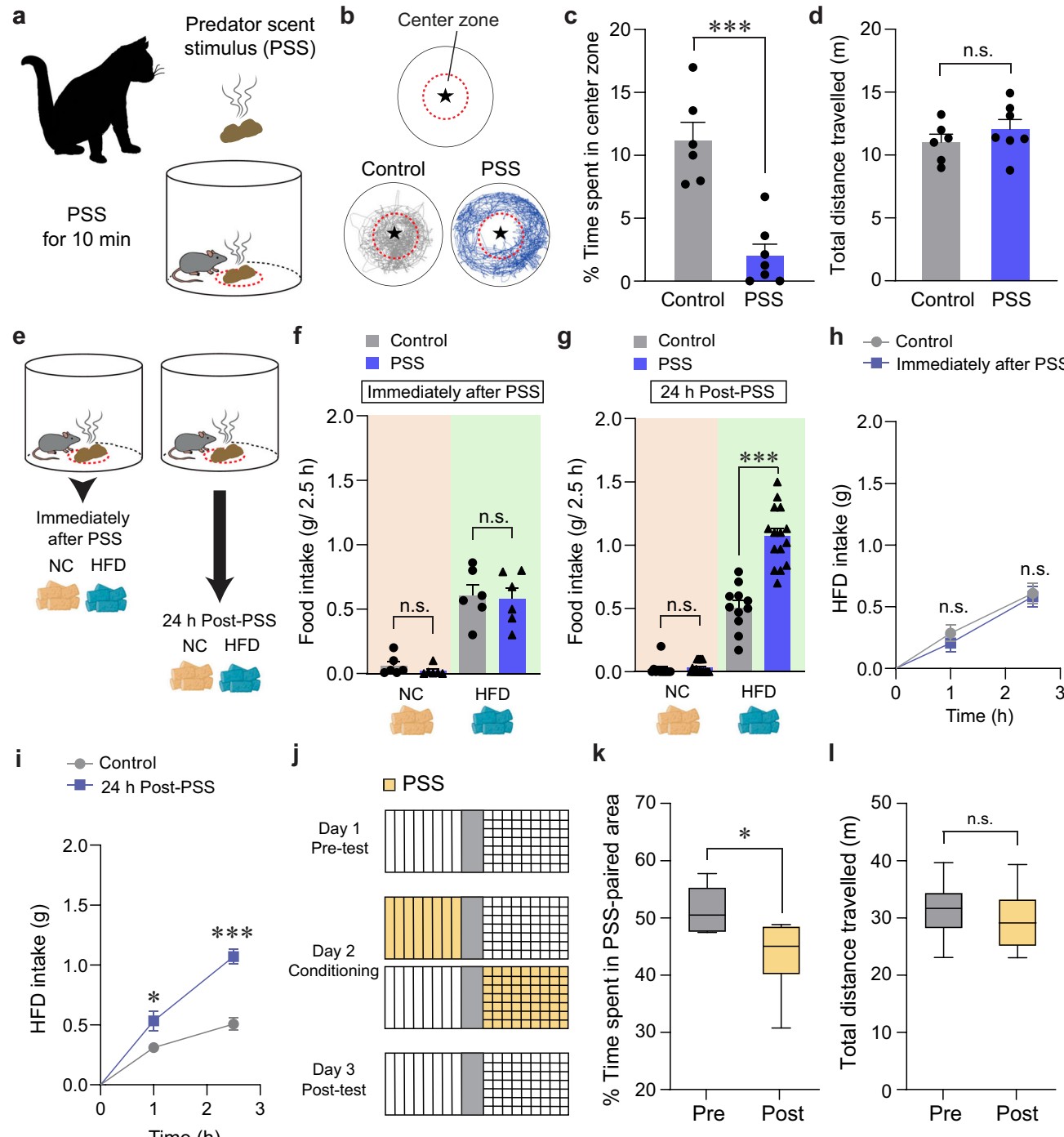

**Fig. 1 | Mice show enhanced food intake for HFD, but not NC, 24 h after PSS exposure. a** Schematics of PSS exposure. **b** Asterisk denotes the location of the small wire mesh cage (top). Representative movement tracks in the presence of control or PSS (bottom). **c, d** Time spent (%) in the center zone and total distance traveled ($n$ = 6, 7 mice per group). In (**c**) two-tailed unpaired $t$ test, $t_{11}$ = 5.491, ***$p$ = 0.000189; In (**d**) two-tailed unpaired $t$ test, $t_{11}$ = −1.013, $p$ = 0.333. **e** Schematic illustrating food consumption test after PSS. **f** 2.5 h food consumption immediately after PSS ($n$ = 6 mice per group). In NC intake, two-tailed unpaired $t$ test, $t_{10}$ = 1.154, $p$ = 0.275; In HFD intake, two-tailed unpaired $t$ test, $t_{10}$ = 0.217, $p$ = 0.832. **g** 2.5 h food consumption 24 h after PSS ($n$ = 11, 15 mice per group). In NC intake, two-tailed unpaired $t$ test, $t_{24}$ = −0.442, $p$ = 0.663; In HFD intake, two-tailed unpaired $t$ test, $t_{24}$ = −6.693, ***$p$ = 0.000000636. **h, i** Cumulative HFD intake immediately (**h**; $n$ = 6

mice per group) or 24 h (**i**; $n$ = 6 mice per group) after PSS exposure. In (**h**) two-way repeated-measures (RM) ANOVA ($F_{(1,10)}$ = 0.292, $p$ = 0.601) was followed by Bonferroni post hoc test for multiple comparisons; $p$ = 0.466, $p$ = 0.814 compared with control mice; In (**i**) two-way RM ANOVA ($F_{(1,10)}$ = 18.390, $p$ = 0.002) was followed by Bonferroni post hoc test for multiple comparisons; *$p$ = 0.035, ***$p$ < 0.001 compared with control mice at 1 h, 2.5 h, respectively. **j** Experimental procedure to test CPA: one side of a two-sided chamber is paired with PSS on day 2. **k, l** Time spent (%) in PSS-paired chambers and total distance traveled ($n$ = 6 mice). Box–whisker plots display median (center) and 2.5–97.5 percentiles of the distribution (bounds) with whiskers extending from min to max values. In (**k**) two-tailed paired $t$ test, $t_5$ = 3.776, *$p$ = 0.0129; In (**l**) two-tailed paired $t$ test, $t_5$ = 0.719, $p$ = 0.504. n.s not significant. Data are expressed as mean ± SEM.

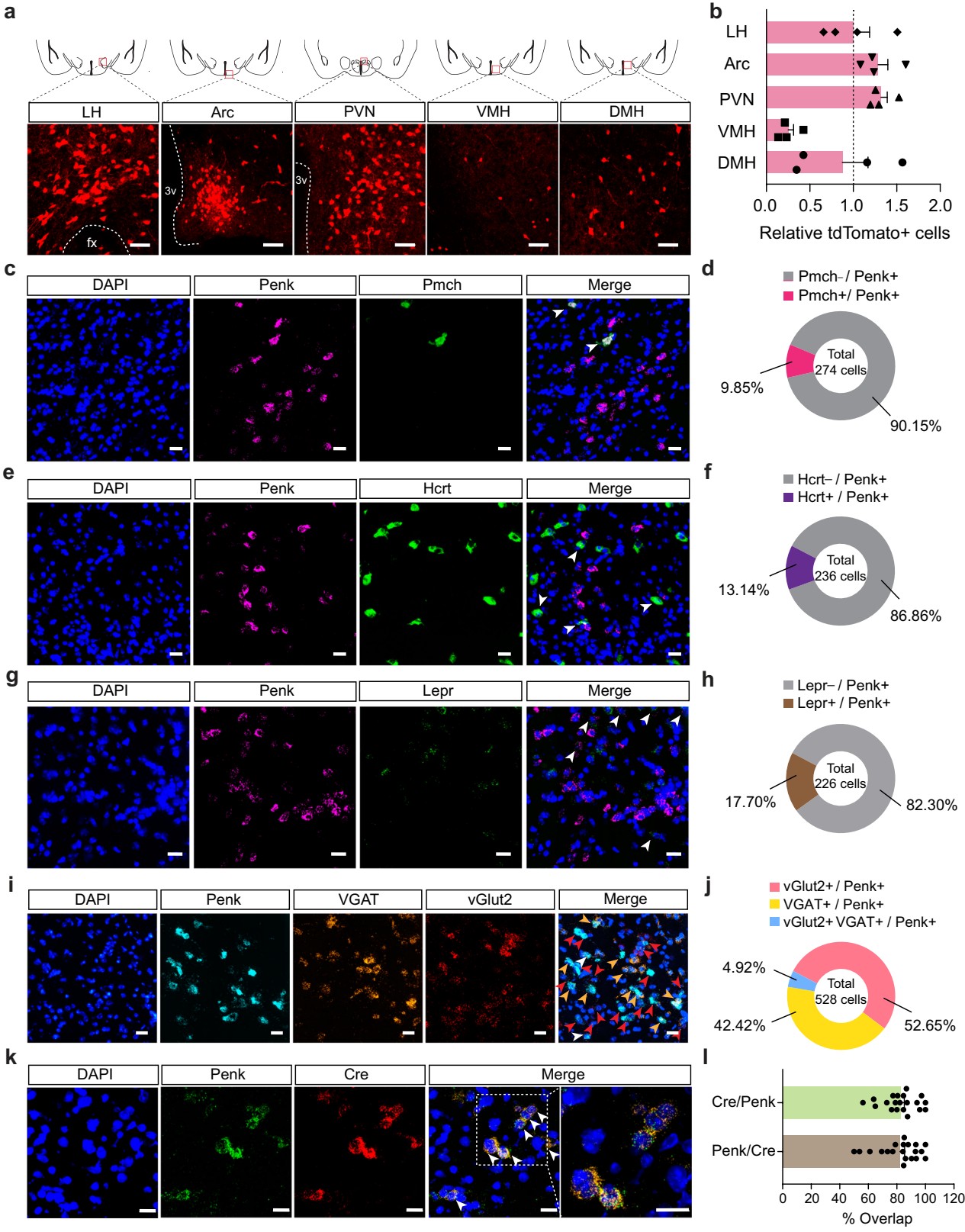

glutamatergic neuronal markers, respectively, we observed that LH[Penk] neurons are composed of subsets of inhibitory and excitatory cells (Fig. 2i, j and Supplementary Fig. 2d).

To gain a circuit-level understanding of how LH[Penk] neurons contribute to PSS-induced HFD overconsumption associated with negative emotional valence, we next aimed to delineate the input organization of the LH[Penk] circuitry. We injected Cre-dependent AAVs

expressing TVA receptor and rabies virus glycoprotein (RVG) (AAV-DIO-mRuby2-TVA, AAV-DIO-RVG, respectively) into the LH of Penk-Cre mice validated to co-express Penk and Cre recombinase (Fig. 2k, l). We then delivered EnvA-pseudotyped, glycoprotein-deleted rabies virus (EnvA-RVΔG-eGFP) into the same area to map monosynaptic inputs to LH[Penk] neurons (Supplementary Fig. 3a–c). We found that LH[Penk] neurons receive synaptic inputs from the medial prefrontal cortex (mPFC),

**Fig. 2 | The LH contains molecularly distinct Penk-expressing neurons.**
**a** Coronal diagrams depicting the region analyzed (squared in red; top) and confocal images showing the Penk expression in the multiple hypothalamic subregions of Penk-Cre × Ai14 mice (bottom), replicated independently with similar results in 4 mice. Scale bars, 50 μm. fx, fornix; 3 v, third ventricle. **b** Quantification of tdTomato-positive cells in the LH, Arc, PVN, VMH, and DMH (n = 4 Penk-Cre × Ai14 mice). Dashed line indicates the normalized level of tdTomato-positive cells in the LH. **c–h**, Representative images of RNA in situ hybridization for Pmch (**c**), Hcrt (**e**), and Lepr (**g**) with Penk in the LH. Arrowheads represent colocalization, replicated independently with similar results in 2 mice. Scale bars, 25 μm. Pie charts indicate % of LH[Penk] neurons colocalizing with Pmch (**d**; n = 274 cells from two mice), Hcrt (**f**; n = 236 cells from two mice), or Lepr (**h**; n = 226 cells from two mice). A large fraction of LH[Penk] neurons does not express Pmch, Hcrt, or Lepr. **i, j** Representative images of RNA in situ hybridization for VGAT and vGlut2 with Penk in the LH (**i**). Red, yellow, and white arrowheads represent colocalization of Penk with vGlut2, VGAT, and all respectively, replicated independently with similar results in 3 mice. Scale bars, 25 μm. Pie charts indicate % of LH[Penk] neurons colocalizing with vGlut2, VGAT and all (**j**; n = 528 cells from three mice). **k, l** Representative images of RNA in situ hybridization for Penk and Cre in the LH using Penk-Cre mice and its high magnification image (squared in white dashed line) (**k**). Arrowheads represent colocalization. Scale bars, 20 μm. Quantification of the % overlap of Cre+ cells as a fraction of Penk+ cells and Penk+ cells as a fraction of Cre+ cells (**l**). (n = 20 images). n.s not significant. Data are presented as mean ± SEM.

nucleus accumbens (NAc), preoptic area (POA), PVN, central amygdala (CeA), and ventral tegmental area (VTA). Among these areas, the PVN -a principal regulator of the HPA axis[33]- is the primary input source of LH[Penk] neurons (Supplementary Fig. 3d, e). To investigate the molecular identity of the PVN input neurons, we combined virus-mediated input tracing with FISH and found that most of the PVN input neurons are glutamatergic (Supplementary Fig. 3f, g). Together, these data indicate that LH[Penk] neurons directly receive information from several brain areas that have been implicated in regulating central stress response systems or in the emotional/behavioral processing of aversive and rewarding stimuli[33–36].

### PSS enhances LH[Penk] neuronal responsiveness upon HFD consumption

Since the specific contributions of the LH[Penk] neurons to mediating PSS-induced emotional overeating remain unknown, we next asked how endogenous LH[Penk] neuronal activity is affected by PSS or HFD exposure. We investigated this by virally expressing Cre-dependent GCaMP6f in the LH of Penk-Cre mice and implanting an imaging cannula with a gradient index (GRIN) lens directly above the virus injection site (Fig. 3a, b). We then acquired single-cell-resolution in vivo Ca²⁺ imaging data from the GCaMP6f-expressing LH[Penk] neurons (Fig. 3c). To determine whether LH[Penk] neuronal activity is associated with PSS-induced aversive emotional responses, we monitored the dynamics of GCaMP6 fluorescence during a 10-min presentation of PSS. We found that when the mice were exposed to PSS, many LH[Penk] neurons showed an increase in the number of Ca²⁺ transients that then returned to baseline after PSS removal (Supplementary Fig. 4a–c). We also observed that more LH[Penk] neurons were activated (41.33%) than inhibited (16.00%) when PSS was presented (Supplementary Fig. 4d–g). In contrast, at the time of PSS removal, only 9.33% of LH[Penk] neurons were activated, but a relatively higher proportion of the cells (32.00%) was inhibited (Supplementary Fig. 4h–k). Notably, in the presence of an empty container with the same color and shape as the one we used for presenting the PSS, the LH[Penk] neurons exhibit no substantial differences in activity (Supplementary Fig. 4l–n). Together, these data suggest that LH[Penk] neuronal activity encodes the PSS-driven negative emotional state rather than general behavioral features evoked by novel stimuli.

We next asked how PSS affects LH[Penk] neurons in the context of palatable food consumption. Based on our observation of emotional overeating 24 h after PSS exposure (Fig. 1g), we examined in vivo Ca²⁺ dynamics of PSS-primed LH[Penk] neurons in response to HFD eating (Fig. 3d). We found increased LH[Penk] neuronal activity in PSS mice upon the HFD eating onset, whereas non-PSS controls showed only a mild increase or no response under the same condition (Fig. 3e–h). Overall, we found that more than 48% of LH[Penk] neurons in PSS mice were activated upon the HFD eating onset compared to 30.91% of LH[Penk] neurons in control mice (Fig. 3i, j). In contrast to this, we found that novel objects evoked only subtle changes in LH[Penk] neuronal activity, regardless of previous PSS exposure (Fig. 3k–r). Together, these data indicate that PSS exposure is a

critical prerequisite for the emergence of activity in LH[Penk] neurons that encodes HFD salience.

We further explored the correlation between PSS-primed LH[Penk] neuronal activity and multiple eating bouts for HFD. After extracting in vivo Ca²⁺ transients from LH[Penk] neurons, we quantified the proportion of Ca²⁺ transients that occurred at the onset of eating bouts. Consistent with our previous data, we found that 76.6% of Ca²⁺ transients from PSS-primed LH[Penk] neurons were generated during HFD consumption (Supplementary Fig. 5d–f). In mice never exposed to PSS, only 38.81% of Ca²⁺ transients responded to the eating bouts (Supplementary Fig. 5a–c). These data support the hypothesis that PSS exposure potentiates the responsiveness of LH[Penk] neurons to HFD eating, which may lead to emotional overeating.

### Modulation of LH[Penk] neuronal activity affects PSS-induced maladaptive behaviors

In light of our previous findings that increased activity of LH[Penk] neurons may encode PSS-induced behavioral changes (Fig. 3, Supplementary Fig. 4), we reasoned that selective activation of the LH[Penk] neurons may recapitulate the behavioral and emotional features of PSS mice, including HFD overconsumption and negative emotional valence. To test this idea, we bilaterally expressed Cre-dependent Gq-coupled designer receptors exclusively activated by designer drugs (DREADDs; hM3D) in the LH of Penk-Cre mice (Fig. 4a). In the presence of clozapine-N-oxide (CNO), an inert ligand specific to the DREADDs[37], the mice expressing hM3D in LH[Penk] neurons showed significantly enhanced consumption of HFD but not NC, whereas the saline-treated mice exhibited no substantial differences in food intake levels (Fig. 4b, c and Supplementary Fig. 6a, b).

Next, we asked whether chemogenetic activation of LH[Penk] neurons can transmit a negative valence signal. Using a CPA protocol (see the Methods; Fig. 4d), we found that mice expressing hM3D in LH[Penk] neurons exhibited a strong aversion to the CNO-paired chamber with regular locomotion (Fig. 4e–g), indicating the aversive nature of LH[Penk] neurons. Analogous results were obtained using a real-time place test (RTPT), with Penk-Cre mice expressing Channelrhodopsin2 (ChR2) in the LH followed by the implantation of an optic cannula in the same site (Supplementary Fig. 6c). Unlike control Penk-Cre mice expressing eYFP alone, the ChR2-expressing Penk-Cre mice avoided the side of the RTPT apparatus paired with photostimulation (10-ms pulses of 473-nm light at 20 Hz) with no accompanying change in locomotion (Supplementary Fig. 6d–f). Moreover, ChR2-mediated activation of LH[Penk] neurons at the same frequency (20 Hz) significantly increased the HFD consumption without affecting NC intake (Supplementary Fig. 6g, h). Together, these data suggest that activation of LH[Penk] neurons increases the tendency of mice to overconsume palatable foods, which is possibly triggered by an aversive or negative emotional state.

These results led us to hypothesize that silencing LH[Penk] neurons could reverse the emotional overeating and negative emotional valence associated with PSS exposure. To suppress LH[Penk] neuronal activity in vivo, we injected the LH of Penk-Cre mice with AAV expressing the Cre-dependent Kir2.1 potassium channel[38]

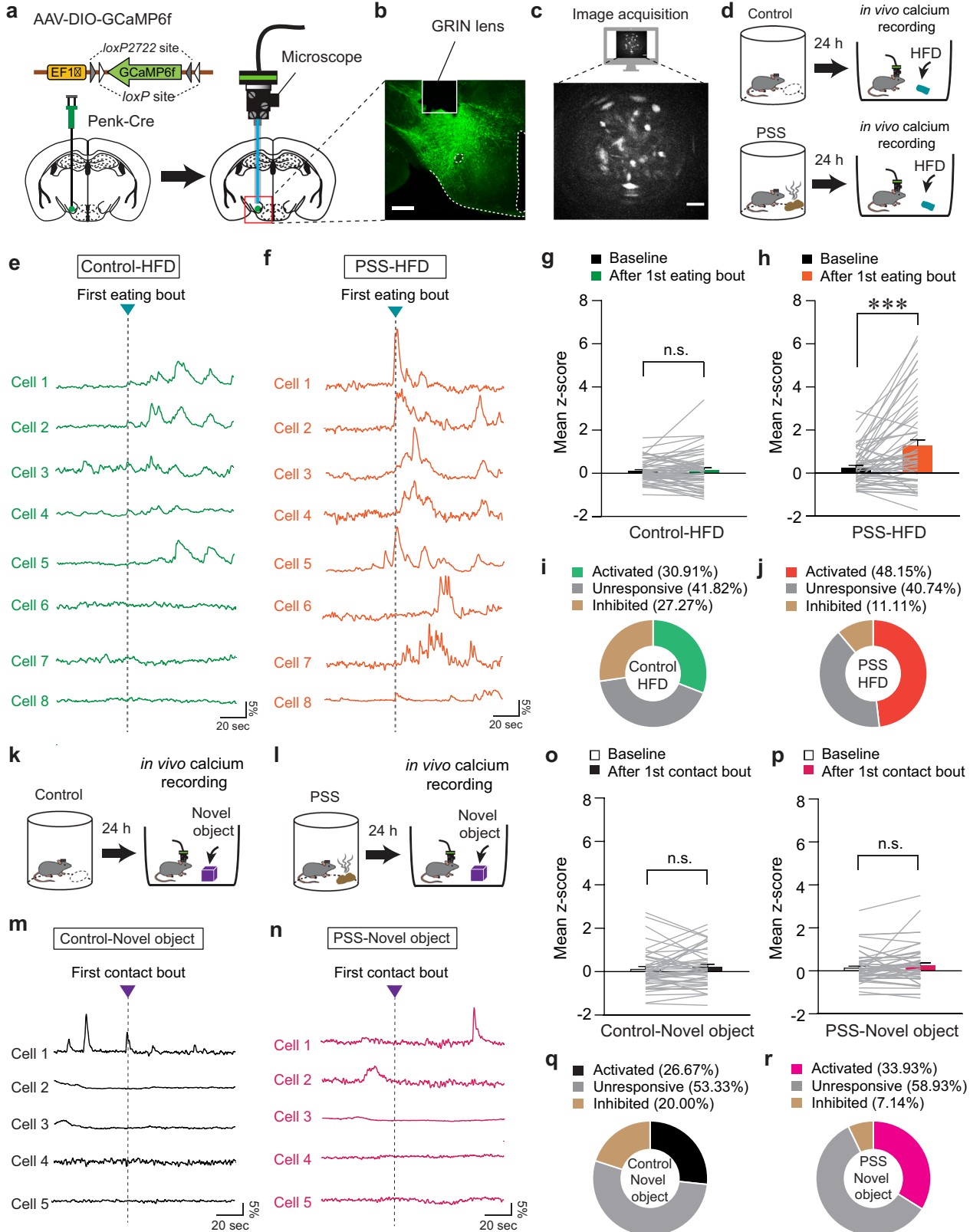

(AAV-DIO-eGFP-Kir2.1) (Fig. 4h). Twenty-four hours after PSS exposure, we confirmed that chronic silencing of LH$^{Penk}$ neurons significantly attenuated HFD overconsumption compared to mice expressing eGFP alone (Fig. 4i). Importantly, this Kir2.1-mediated LH$^{Penk}$ neuronal inhibition did not affect NC intake (Fig. 4j) or cause any accompanying changes in locomotion or olfactory perception (Supplementary Fig. 6i–k). These data suggest that inhibition of

LH$^{Penk}$ neurons can alleviate the palatable food overconsumption triggered by PSS.

We further investigated whether LH$^{Penk}$ neuronal inhibition is sufficient to prevent the negative emotional states that follow PSS exposure. We subjected mice expressing Kir2.1 in the LH$^{Penk}$ neurons to a CPA paradigm paired with PSS and monitored their behaviors (Supplementary Fig. 6l). We found that while eYFP-expressing controls

**Fig. 3 | LH^Penk neurons of PSS mice exhibit sensitized activity in response to HFD consumption. a** Schematic for the injection of AAV-DIO-GCaMP6f (left) and implantation of GRIN lens into the LH (right). **b, c** Confocal image showing GRIN lens placement on the GCaMP6f-expressing LH^Penk neurons. Scale bar, 250 μm (**b**). A sample image for data acquisition. Scale bar, 50 μm (**c**), replicated independently with similar results in 4 mice. **d** Schematic for GRIN lens-implanted mice exposed to HFD. **e, f** Example in vivo Ca²⁺ activity traces from LH^Penk neurons of control (**e**) and PSS mice (**f**) in response to the 1st HFD eating bout. **g, h** Mean z-score of the LH^Penk neuronal activity of control (**g**; $n = 55$ cells from four control Penk-Cre mice) and PSS mice (**h**; $n = 54$ cells from five PSS Penk-Cre mice) before and after the 1st HFD eating bout. Two-tailed paired $t$ test; in (**g**) $t_{54} = -0.435$, $p = 0.665$; in (**h**),

$t_{53} = -4.233$, ***$p = 0.0000921$. **i, j** Pie charts indicate the classification of LH^Penk neurons of control (**i**) and PSS mice (**j**) showing activated, inhibited, or unresponsive activity upon the 1st HFD eating bout. **k, l** Schematic for GRIN lens-implanted control (**k**) and PSS mice (**l**) in the presence of a novel object. In vivo Ca²⁺ activity is monitored in the presence of a novel object 24 h after PSS exposure. **m, n** As for (**e, f**), but in response to the 1st physical contact to a novel object. **o, p** Mean z-score of the LH^Penk neuronal activity of control (**o**; $n = 45$ cells from four control Penk-Cre mice) and PSS mice (**p**; $n = 56$ cells from five PSS Penk-Cre mice) before and after the 1st physical contact to a novel object. Two-tailed paired $t$-test; in (**o**), $t_{44} = -1.036$, $p = 0.306$; in (**p**), $t_{55} = -1.602$, $p = 0.115$. **q, r** As for (**i, j**), but upon the 1st physical contact with a novel object. n.s not significant. Data are presented as mean ± SEM.

spent less time in the PSS-paired chamber, mice expressing Kir2.1 did not show avoidance behaviors (Supplementary Fig. 6m–o). Together with these results, we speculated that Kir2.1-mediated inhibition of LH^Penk neurons attenuates the palatable food overconsumption associated with a negative emotional state that we observed in PSS mice.

## Roles of enkephalinergic systems in the LH circuitry on PSS-induced maladaptive changes

Enkephalins are produced from a propeptide precursor, Penk mRNA, which encodes distinct form of endogenous opioid peptides[39]. We therefore investigated the functional roles of endogenous enkephalin in the LH and the related circuit elements in mediating the effects of PSS on emotional eating. Given previous studies indicating emerging roles of enkephalin signaling in adaptation to stressful experiences[40], we asked whether the Penk in the LH is necessary for HFD overconsumption after PSS exposure. To test this, we bilaterally injected AAV carrying a short hairpin RNA (shRNA) against Penk in a Cre-dependent manner (AAV-DIO-EmGFP-Penk shRNA) into the LH of Penk-Cre mice and confirmed a significant reduction of Penk mRNA expression without accompanying changes in the Lepr, Hcrt and Pmch (Fig. 5a, b). At 24 h post-PSS exposure, we found that shRNA-mediated knockdown of Penk significantly reduced HFD consumption, but not NC intake, whereas PSS mice injected with control virus (AAV-DIO-eGFP) showed HFD overconsumption (Fig. 5c–e). These results indicate that downregulation of Penk in the LH has a critical role in the protection against PSS-induced HFD overconsumption.

Because our previous data showed PSS mice displaying enhanced expression of c-fos in the LH in response to HFD eating (Supplementary Fig. 1a, b), we hypothesized that enkephalin-mediated signaling is necessary to potentiate the responsiveness of PSS-primed LH^Penk neurons during emotional overeating. We first asked whether the HFD-induced expression of c-fos observed in PSS mice mainly occurs in the LH^Penk neurons compared to Lepr-expressing LH (LH^Lepr) neurons representing mostly distinct subpopulation from the LH^Penk neurons (Fig. 2h). We injected AAV-DIO-eGFP into the LH of Penk-Cre or Lepr-Cre mice to visualize LH^Penk and LH^Lepr neuronal cell bodies, respectively, and then measured the proportion of c-fos-positive LH cells 24 h after PSS exposure. In response to HFD, we found enhanced c-fos induction in the LH^Penk neuronal population by up to 29.91%, but to a lesser extent, only 10.72% of c-fos expression were overlapped with eGFP-expressing LH^Lepr cells (Fig. 5f, g). Under the same condition, the total number of c-fos-positive cells in the overall LH was increased consistently in both groups, regardless of cell-type specific eGFP-labeling (Fig. 5h). These data suggest that LH^Penk neurons constitute a more relevant population than LH^Lepr neurons in controlling PSS-induced HFD overconsumption. Importantly, shRNA-mediated knockdown of Penk significantly decreased HFD-induced c-fos expression in the PSS-primed LH, particularly within the LH^Penk neurons (Fig. 5f–h), confirming that the enkephalin peptide plays a critical role in increasing LH^Penk neuronal reactivity to HFD, which is a hallmark of PSS mice exhibiting emotional overconsumption.

To further confirm the necessity of endogenous enkephalin in the LH, we used CRISPR-SaCas9 viral-based system by bilaterally

injecting AAV-CMV-FLEX-SaCas9-U6-sgPenk[18] into the LH of Penk-Cre mice (Fig. 5i). Similar to the effects of shRNA-mediated knockdown of Penk, CRISPR infection reduced Penk expression by 70% (Fig. 5j) and behaviorally suppressed PSS-induced HFD overconsumption without changing NC intake (Fig. 5k, l). Furthermore, in the CPA paradigm, CRISPR-mediated reduction of Penk in the LH resulted in decreasing behavioral avoidance for PSS-paired chamber (Fig. 5m, n), suggesting that LH endogenous enkephalin is necessary for mediating PSS-induced maladaptive behavioral responses including HFD overconsumption and associated negative emotional state.

Because enkephalins are potent endogenous agonists binding to MORs, we next asked whether MORs in the LH^Penk neuronal efferent projections are the receptor targets that relay the signals from the enkephalin-containing LH neurons. To delineate the efferent projections of LH^Penk neurons, we injected AAV-DIO-eGFP into the LH of Penk-Cre mice and found that LH^Penk neurons send projections to the ventral pallidum (VP), VTA, substantia nigra (SN), periaqueductal gray (PAG), and lateral parabrachial nucleus (LPBN) (Fig. 6a–c). Among these labeled regions, the PAG, one of the brain regions thought to be responsible for directing the consumption of innately palatable substances[41,42], receives a dense LH^Penk axonal projection particularly into the lateral part (LPAG). Since MORs are highly expressed in the same area[43], we reasoned that MORs in the LPAG are necessary for triggering HFD overconsumption after artificial activation of LH^Penk neurons. We tested this idea by chemogenetically activating LH^Penk neurons and simultaneously inhibiting MORs through intracerebral microinjections of the MOR-selective antagonist CTAP (0.1 or 1 μg per side) into the LPAG (Fig. 6d, e). Consistent with our previous findings (Fig. 4), CNO-mediated activation of LH^Penk neurons significantly increased HFD consumption (g/2.5 h). However, a local infusion of CTAP into the LPAG, particularly at a high dose (1 μg per side), strongly blocked the HFD consumption even in the presence of CNO, while mice in all groups showed regular NC intake (Fig. 6f, g). Notably, a microinfusion of CTAP (1 μg per side) into the LPAG did not affect general locomotion (Fig. 6h), suggesting that the pharmacological inhibition of MORs in the LPAG specifically suppresses HFD eating behaviors driven by chemogenetic activation of LH^Penk neurons.

Because activation of LH^Penk neurons is a critical component to mediate PSS-induced HFD overconsumption (Fig. 4), we also asked whether the pharmacological inhibition of MORs in the LPAG can normalize maladaptive behavioral responses developed after PSS exposure. Indeed, at 24 h post-PSS exposure, intracranial microinfusion of CTAP (1 μg per side) into the LPAG prevented PSS mice from showing augmented eating upon HFD exposure (Fig. 6i, j). In contrast to this, the same pharmacological inhibition of MORs in the LPAG did not affect aversive behavioral responses for PSS-paired compartment in the CPA procedure (Fig. 6k–n). These data suggest that the activation of MOR signaling in the LPAG is preferentially required for eliciting PSS-induced HFD overconsumption, but not negative emotional state. Together, these results support that endogenous enkephalinergic system (e.g., enkephalin peptides and their receptors) in the LH circuitry is important for regulating the tendency to develop palatable

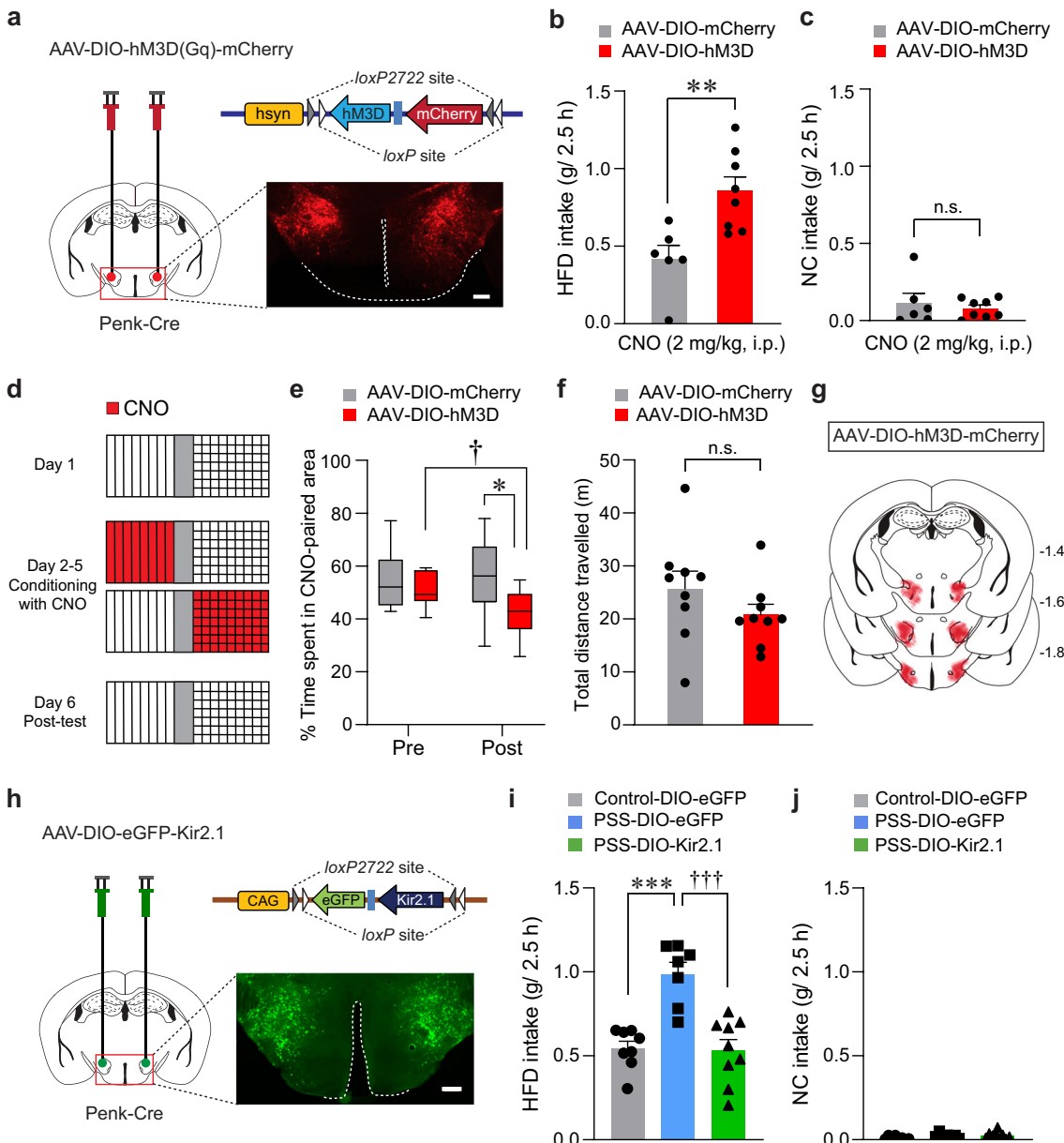

**Fig. 4 | Modulation of LH^Penk neuronal activity affects PSS-induced maladaptive behaviors. a** Cre-dependent AAV expressing hM3D was injected in the LH of Penk-Cre mice, replicated independently with similar results in 4 mice. Scale bar, 250 µm. **b, c** CNO-induced activation of LH^Penk neurons potentiates consumption of HFD, but not NC ($n = 6$, 8 mice for each group). In (**b**), two-tailed unpaired $t$ test, $t_{12} = -3.374$, **$p = 0.00552$; In (**c**), two-tailed unpaired $t$ test, $t_{12} = 0.582$, $p = 0.571$. **d** Experimental procedure to test CPA: one side of a two-sided chamber is paired with CNO injections (2 mg/kg, i.p) during day 2–5. **e** Time spent (%) in CNO-paired side of mice expressing DIO-mCherry or DIO-hM3D during pre- and post-test ($n = 9$ mice per group). Box–whisker plots display median (center) and 2.5 to 97.5 percentiles of the distribution (bounds) with whiskers extending from min to max values. Two-way RM ANOVA ($F_{(1,16)} = 5.011$, $p = 0.04$) was followed by Bonferroni post hoc test for multiple comparisons; *$p = 0.015$ for mice with mCherry vs. hM3D

during post-test; †$p = 0.016$ for hM3D mice during pre- vs. post-test. **f** Total distance traveled during the post-test of CPA experiment in (**e**) ($n = 9$ mice per group). Two-tailed unpaired $t$ test, $t_{11} = 1.660$, $p = 0.125$. **g** Summary diagram for the coverage of hM3D-mCherry viral infusion in the LH of Penk-Cre mice in (**a**–**f**). **h** Cre-dependent AAV expressing Kir2.1 was injected in the LH of Penk-Cre mice, replicated independently with similar results in 5 mice. Scale bar, 250 µm. **i, j** Kir2.1-mediated inhibition of LH^Penk neurons of PSS mice reduces the consumption of HFD, but not NC ($n = 8$, 7, and 9 mice per group). In (**i**), one-way ANOVA ($F_{(2,21)} = 17.730$, $p < 0.001$) was followed by Fisher LSD post hoc test for multiple comparisons; ***$p < 0.001$ compared with DIO-eGFP-expressing control mice; †††$p < 0.001$ compared with DIO-eGFP-expressing PSS mice. n.s not significant. Data are presented as mean ± SEM.

food overconsumption, substantiating the unique role of LH^Penk neurons in mediating post-PSS effects.

## CORT mimics the delayed onset of emotional overeating after PSS

The HPA system plays a primary role in coordinating endocrine and behavioral responses to stress. Activation of the HPA axis results in

the release of corticotropin-releasing hormone (CRH) from the PVN. This, in turn, increases glucocorticoid hormone levels (primarily corticosterone in rodents and cortisol in humans), which can cause lasting changes in neuronal function and behaviors[23,44,45]. Notably, we found that while serum CORT levels are significantly elevated 30 min after PSS exposure, they return to baseline 24 h later (Fig. 7a, b).

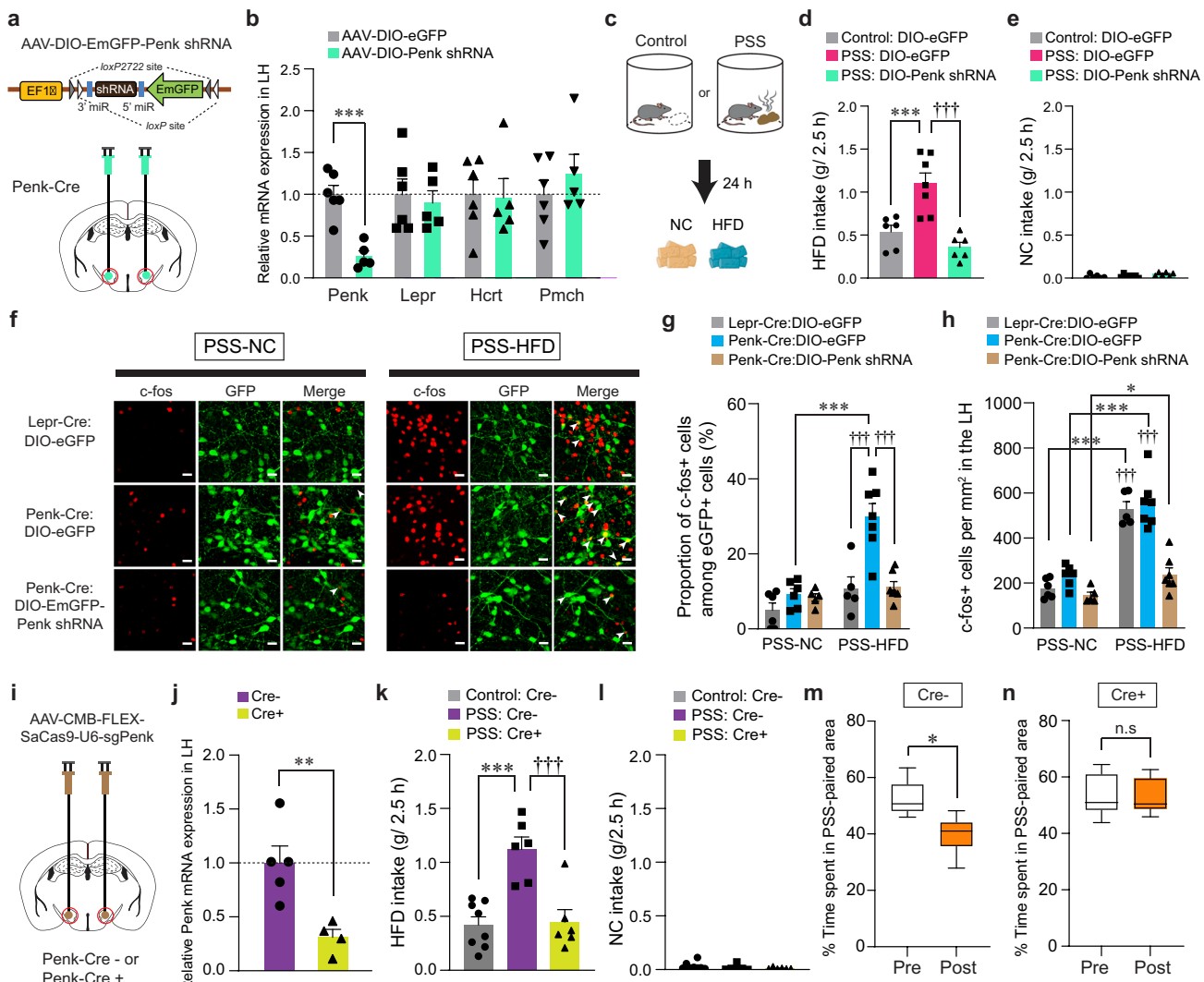

**Fig. 5 | Endogenous enkephalin in the LH is necessary for PSS-induced maladaptive responses. a, b** Schematic for the injection of AAV-DIO-Penk shRNA into the LH (**a**). Red circles denote the dissected regions used in RT-qPCR (**b**; $n = 6, 5$ mice per group). In (**b**), two-tailed unpaired $t$ test, $t_9 = 5.589$, ***$p = 0.000339$. **c–e** 2.5 h food consumption test ($n = 6, 7, 6$ mice per group). In (**d**), one-way ANOVA ($F_{(2,16)} = 16.713$, $p < 0.001$) was followed by Fisher LSD post hoc test for multiple comparisons; ***$p < 0.001$, †††$p < 0.001$ compared with eGFP-expressing PSS mice. **f** Representative images showing c-fos immunoreactivity and eGFP in the LH of Lepr-Cre and Penk-Cre mice. Arrowheads indicate the colocalization of c-fos with eGFP. Scale bars, 20 μm. **g, h** Quantification of the proportion of c-fos+ cells among eGFP-labeled LH neurons (**g**) or c-fos+ cells in the LH (**h**) ($n = 6, 6, 5$ mice for each PSS-NC group; $n = 5, 7, 7$ mice for each PSS-HFD group). In (**g**), two-way ANOVA ($F_{(2,30)} = 8.543$, $p = 0.001$) was followed by Fisher LSD post hoc test for multiple comparisons; ***$p < 0.001$, †††$p < 0.001$ compared with Penk-Cre: DIO-eGFP at PSS-

HFD. In (**h**), two-way ANOVA ($F_{(2,30)} = 11.261$, $p < 0.001$) was followed by Fisher LSD post hoc test for multiple comparisons; *$p = 0.034$, ***$p < 0.001$ compared with mice at respective PSS-HFD conditions; †††$p < 0.001$ compared with Penk-Cre: DIO-Penk shRNA at PSS-HFD. **i, j** Schematic for CRISPR virus injections into the LH (**i**). Red circles denote the dissected LH used in RT-qPCR (**j**; $n = 5, 4$ mice per group). In (**j**), two-tailed unpaired $t$ test, $t_7 = 3.631$, **$p = 0.00839$. **k, l** 2.5 h food consumption test ($n = 8, 6, 6$ mice per group). In (**k**), one-way ANOVA ($F_{(2,17)} = 15.477$, $p < 0.001$) was followed by Fisher LSD post hoc test for multiple comparisons; ***$p < 0.001$, †††$p < 0.001$ compared with PSS: Cre-. **m, n** Time spent (%) in PSS-paired side of mice receiving CRISPR virus in the LH ($n = 6$ mice). Box–whisker plots display median (center) and 2.5 to 97.5 percentiles of the distribution (bounds) with whiskers extending from min to max values. In (**m**), two-tailed paired $t$ test, $t_5 = 3.892$, *$p = 0.0115$; In (**n**), two-tailed paired $t$ test, $t_5 = 0.186$, $p = 0.86$. n.s not significant. Data are expressed as mean ± SEM.

To understand whether CORT promotes emotional overeating via directly acting on LH^Penk neurons, we first asked whether LH^Penk neurons express the CORT receptors such as GR or mineralocorticoid receptor (MR)[46]. We found that most LH^Penk neurons expressed GR (79.48%) rather than MR (Fig. 7c, d, and Supplementary Fig. 7a), suggesting that a substantial fraction of LH^Penk neurons can convey CORT signals through GR.

We next sought to determine whether mice treated with CORT exhibit HFD overconsumption previously seen in PSS mice. Immediately after CORT administration (2 mg/kg, s.c.), we did not observe any changes in HFD or NC intake. Twenty-four hours after CORT injection, however, mice showed HFD overconsumption without any change in

NC intake (Supplementary Fig. 7b, c), recapitulating the PSS-driven delayed onset of overeating. Since most LH^Penk neurons express GR (Fig. 7c), we hypothesized that GR activation in the LH is required for the CORT-induced HFD overconsumption. To test this hypothesis, we performed an intracranial microinfusion of the GR antagonist, mifepristone (1 or 10 ng per side) into the LH followed by systemic injections of vehicle or CORT (Fig. 7e, f, and Supplementary Fig. 7d). On the next day, we observed that mifepristone microinfusion prevented the CORT pretreatment from increasing HFD intake in a dose-dependent manner with no accompanying changes in NC intake (Fig. 7g, h). We did not observe any differences in locomotion of the mifepristone-microinjected mice (Supplementary Fig. 7e), suggesting the

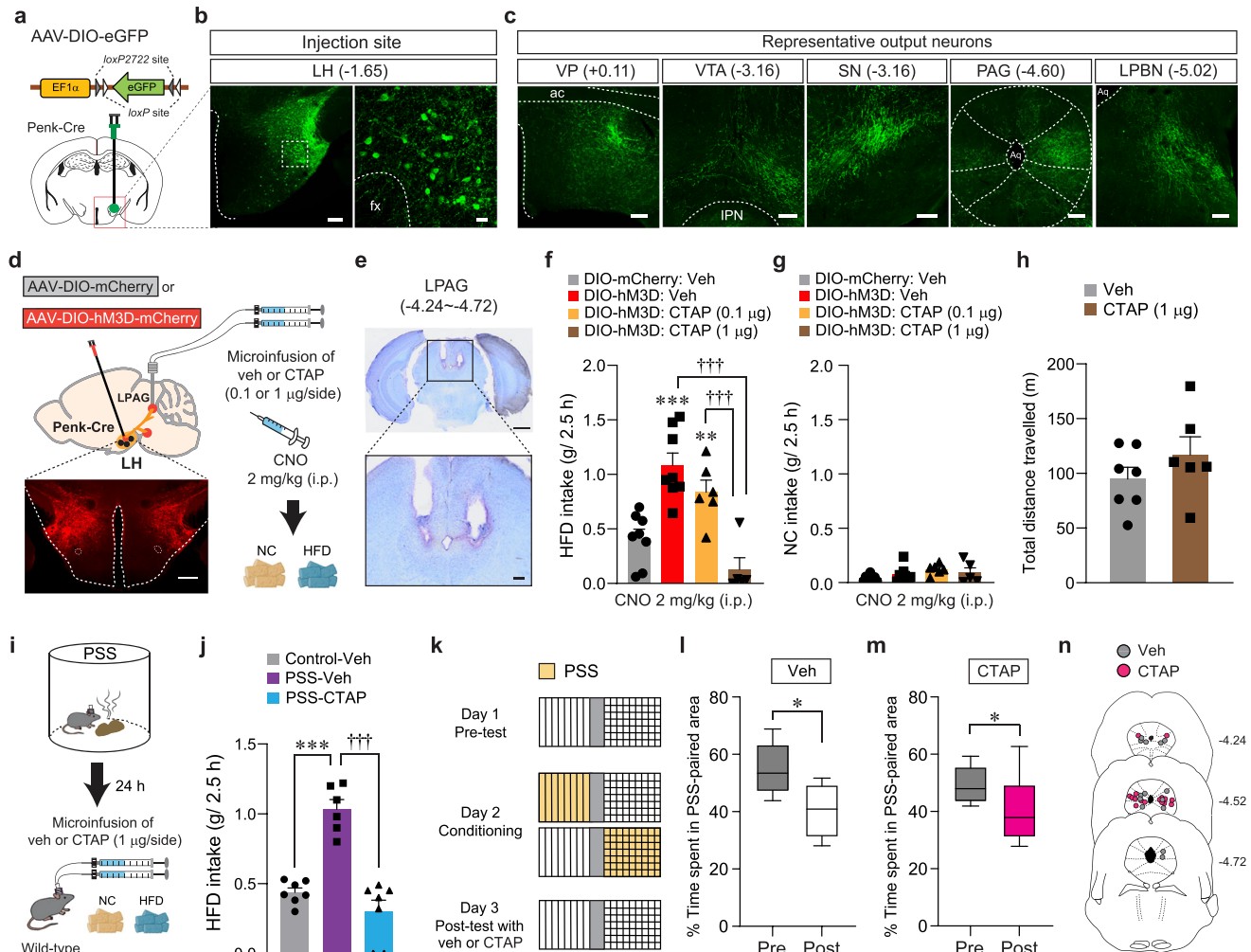

**Fig. 6 | Pharmacological inhibition of MORs in the LPAG reduces HFD consumption. a** Schematics for the injection of AAV-DIO-eGFP into the LH of Penk-Cre mice. **b** Images showing eGFP expression in the LH[Penk] neuronal cells (left), and the high magnification image of a region in the white dashed box (right), replicated independently with similar results in 3 mice. Scale bar, 200 µm, 25 µm, respectively. fx, fornix. **c** eGFP-positive axons from LH[Penk] neuron were found in the VP, VTA, SN, PAG, and LPBN, replicated independently with similar results in 3 mice. Scale bars, 200 µm. ac, anterior commissure; IPN, interpeduncular nucleus; Aq, aqueduct. **d** Schematic for microinjections of CTAP into the LPAG of mice expressing mCherry or hM3D in the LH[Penk] neurons. Confocal image showing hM3D-expressing LH[Penk] neurons, replicated independently with similar results in 5 mice. Scale bar, 500 µm. **e** Images show the location of the cannula tips in the LPAG, replicated independently with similar results in 5 mice. Scale bars, 1 mm (top) and 200 µm (bottom). **f, g** 2.5 h food consumption ($n = 8, 8, 6, 5$ mice per

group). In (**f**), one-way ANOVA ($F_{(3,23)} = 15.946$, $p < 0.001$) was followed by Fisher LSD post hoc test for multiple comparisons; **$p = 0.008$, ***$p < 0.001$ compared with vehicle-mCherry mice; †††$p < 0.001$ compared with CTAP (1 µg)-hM3D mice. **h** Locomotor activity ($n = 7, 6$ mice per group). **i** Schematic for microinjection of veh or CTAP into the LPAG of PSS mice before HFD exposure. **j** 2.5 h HFD consumption test ($n = 7, 6, 7$ mice per group). One-way ANOVA ($F_{(2,17)} = 35.847$, $p < 0.001$) was followed by Fisher LSD post hoc test for multiple comparisons; ***$p < 0.001$, †††$p < 0.001$ compared with PSS mice received vehicle. **k–m** Time spent (%) in the PSS-paired chambers ($n = 7, 8$ mice per group). Box–whisker plots display median and 2.5 to 97.5 percentiles of the distribution with whiskers extending from min to max values. In (**l**), two-tailed paired $t$ test, $t_6 = 2.776$, *$p = 0.0322$. In (**m**), two-tailed paired $t$ test, $t_7 = 2.978$, *$p = 0.0206$. **n** Locations of the cannula tips in the mice included in (**l, m**). Data are expressed as mean ± SEM.

normalization of HFD overconsumption we observed with mifepristone microinfusion cannot be ascribed to altered movements. Together, these data indicate that activation of GR signaling in the LH is necessary for mediating the delayed onset of HFD overconsumption after CORT treatments.

To further confirm the active role of the GR specifically expressed in the LH[Penk] neurons, we asked whether LH[Penk] neuron-specific overexpression of GR can reverse the effect of mifepristone in the LH of CORT-treated mice. We stereotaxically injected AAV expressing GR in a Cre-dependent manner (AAV-hsyn-DIO-GR) into the LH of Penk-Cre mice, followed by implantation of drug-infusion cannulae above the virus injected site (Fig. 7i). We confirmed that GR mRNA levels were robustly increased both in FISH experiments and quantitative real-time PCR (qRT-PCR) assay (Fig. 7j and Supplementary Fig. 7f). We

also found that the specific overexpression of GR in the LH[Penk] neurons blocked the mifepristone-induced normalization of HFD overconsumption in CORT-treated mice (Fig. 7k), suggesting that inhibition of GR signaling in the LH[Penk] neurons is a critical component for mifepristone to suppress the effects of CORT on triggering palatable food overconsumption.

To determine how CORT regulates the basal activity of LH[Penk] neurons, we next virally expressed Cre-dependent GCaMP6f in the LH of Penk-Cre mice and monitored in vivo Ca²⁺ dynamics following systemic CORT injection (2 mg/kg, s.c.). Five minutes after CORT administration, we observed a decrease in the average number of Ca²⁺ transients per minute in LH[Penk] neurons, which was recovered shortly to the level of one in vehicle-treated animals (Supplementary Fig. 7g, h). Twenty-four hours after CORT treatment, we no longer observed

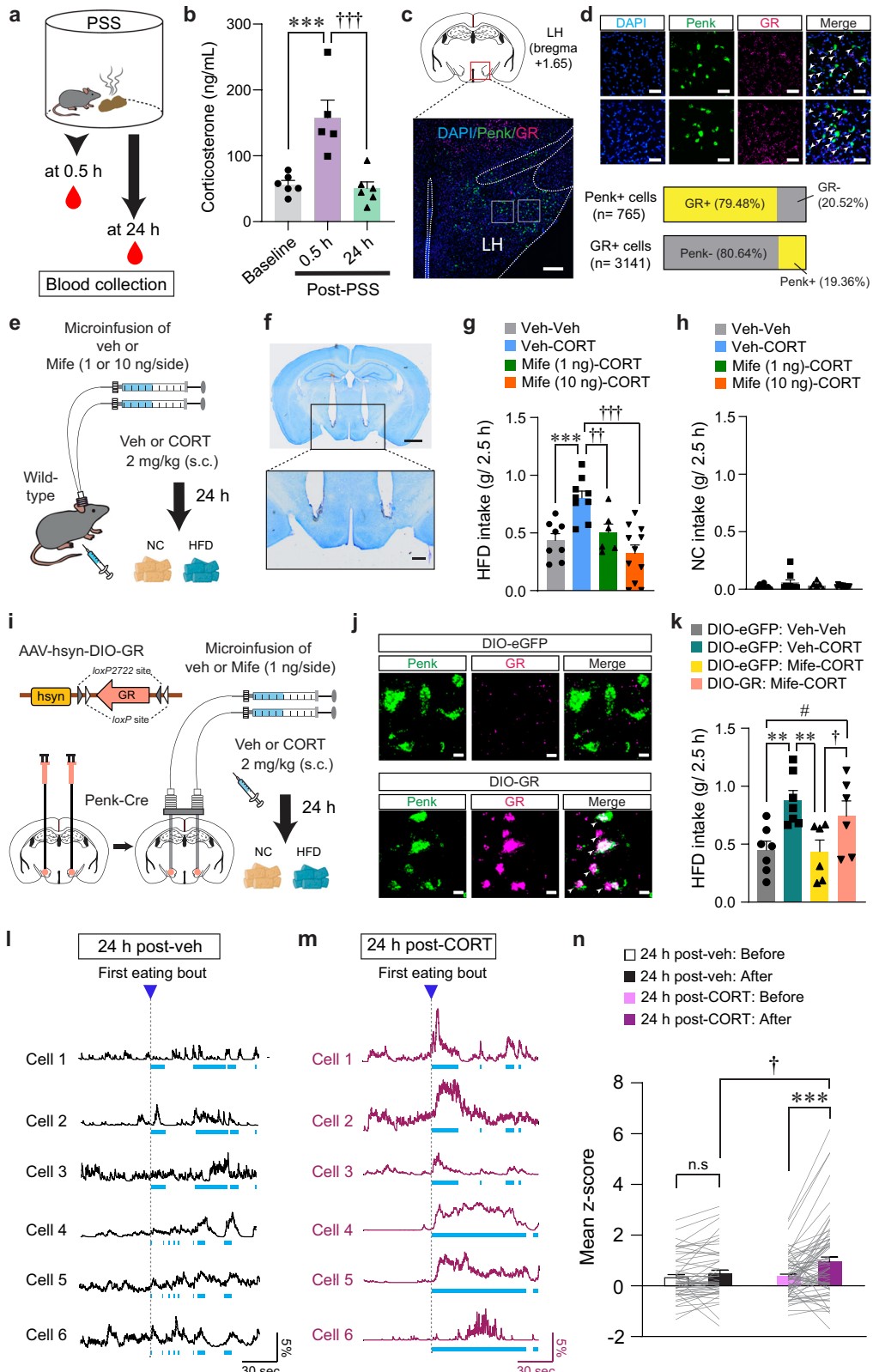

any difference in the baseline activity of the LH[Penk] neurons (Supplementary Fig. 7i–l). These data demonstrate that acute CORT treatment induces a transient reduction in baseline LH[Penk] neuronal activity in vivo, which was followed by a rapid recovery.

We next explored how CORT pretreatment affects the in vivo Ca[2+] dynamics of LH[Penk] neurons during HFD consumption. Twenty-four hours after drug administration, we found that mice treated with CORT showed higher LH[Penk] neuronal reactivity at the first onset of HFD eating bout compared to vehicle-treated mice (Fig. 7l–n). Furthermore, we investigated whether CORT pretreatment increases the HFD-induced expression of c-fos in the LH[Penk] neurons. With CORT pretreatment, HFD increased the proportion of c-fos−positive LH[Penk] neurons up to 31.6%, but vehicle-treated mice exhibited only 7.4% and 11.16% of LH[Penk] neurons c-fos−positive upon NC or HFD exposure,

**Fig. 7 | CORT increases HFD intake and LH^Penk neuronal reactivity to HFD consumption. a, b** Serum CORT levels after PSS (**b**; n = 6, 5, and 6 mice). In (**b**), One-way ANOVA ($F_{(2,14)}$ = 14.634, $p$ < 0.001) was followed by Fisher LSD post hoc test for multiple comparisons; ***$p$ < 0.001, †††$p$ < 0.001 compared with mice at 0.5 h. **c** Representative image of FISH for Penk and GR in the LH, replicated independently with similar results in 3 mice. Scale bars, 250 μm. **d** Enlarged views of regions in the white boxes showing in (**c**), replicated independently with similar results in 3 mice. Scale bars, 50 μm. Arrowheads represent colocalization (top). Bar charts indicate % of Penk+ or GR + LH neurons colocalizing with GR or Penk, respectively (bottom). **e** Schematic for the microinjection of vehicle or mifepristone into the LH followed by systemic injections. **f** Images show the location of the cannula tips in the LH, replicated independently with similar results in 6 mice. Scale bars, 1 mm (top) and 250 μm (bottom). **g, h** 2.5 h food consumption (n = 8, 9, 6, and 11 mice). In (**g**), one-way ANOVA ($F_{(3,30)}$ = 10.336, $p$ < 0.001) was followed by Fisher LSD post hoc test for multiple comparisons; ***$p$ < 0.001 compared with mice received veh only; ††$p$ = 0.007, †††$p$ < 0.001 compared with mice received veh (μ-injection) and CORT

(s.c.). **i** Schematic for the bilateral injection of AAV-DIO-GR into the LH, followed by implantation of cannula. **j** Representative images of FISH for Penk and GR in the LH of Penk-Cre mice expressing DIO-eGFP (top) and DIO-GR (bottom). Arrowheads represent colocalization, replicated independently with similar results in 2 mice. Scale bars, 10 μm. **k** 2.5 h HFD consumption test (n = 7, 7, 6, and 6 mice). One-way ANOVA ($F_{(3,22)}$ = 5.149, $p$ = 0.008) was followed by Fisher LSD post hoc test for multiple comparisons; **$p$ = 0.004, **$p$ = 0.004 compared with DIO-eGFP mice received veh (μ-injection) and CORT (s.c.); †$p$ = 0.045, #$p$ = 0.047 compared with DIO-GR mice received Mife (μ-injection) and CORT (s.c.). **l, m** Example in vivo Ca²⁺ traces in response to HFD eating. Light blue dashed areas indicate eating bouts. **n** Mean z-score of the LH^Penk neuronal activity in veh- (n = 51 cells from 5 mice) or CORT-treated mice (n = 68 cells from 5 mice). Two-way RM ANOVA ($F_{(1,116)}$ = 6.141, $p$ = 0.015) was followed by Bonferroni post hoc test for multiple comparisons; ***$p$ < 0.001, †$p$ = 0.021 compared with CORT-treated mice after 1st HFD eating bout; $p$ = 0.27 for veh-treated mice before vs. after 1st HFD eating bout. n.s not significant. Data are presented as mean ± SEM.

respectively (Supplementary Fig. 8a–c). Moreover, a local infusion of mifepristone into the LH prior to CORT administration led to significant reduction in c-fos–positive LH^Penk neurons (5.48%) after HFD exposure, confirming that the HFD-induced c-fos expression was mediated by GR activation in the LH after CORT treatment (Supplementary Fig. 8d–f). Together, these results support that CORT pretreatment can increase the responsiveness of the LH^Penk neurons to HFD eating, which is a hallmark previously seen in PSS mice (Fig. 3).

### Inhibition of GR in the LH reverses PSS-induced maladaptive changes

Our results led us to hypothesize that PSS-induced CORT elevation may alter LH^Penk neuronal responsiveness to HFD via activation of GR signaling, leading to emotional overeating. We therefore asked whether pharmacological inhibition of GR signaling in the LH normalizes the maladaptive eating and negative emotional behaviors that follow PSS exposure. We performed an intracranial microinfusion of either vehicle or mifepristone (1 or 10 ng per side) into the LH before PSS exposure (Fig. 8a, b, and e). Twenty-four hours later, we found that PSS mice that received mifepristone showed a dose-dependent decrease in HFD consumption (Fig. 8c). Regardless of drug treatment, we did not observe changes in NC consumption in any of the groups (Fig. 8d). These results suggest GR activation in the LH is necessary for PSS-induced HFD overconsumption.

To determine whether pharmacological inhibition of GR in the LH blocks the negative emotional state that follows PSS exposure, we performed a place preference conditioning test. We delivered mifepristone into the LH of mice before subjecting them to a PSS-paired side chamber on day 2 (Fig. 8f). On test day, we found that the local infusion of mifepristone alleviated the PSS-induced post-conditioning avoidance, whereas vehicle-treated mice spent less time in the chamber paired with PSS (Fig. 8g, h). Together, these data suggest that GR signaling inhibition in the LH can normalize the HFD overconsumption and aversive emotional state induced by PSS exposure.

Given our previous data showing that increased LH^Penk neuronal activity is likely to encode PSS-induced HFD overconsumption (Figs. 3 and 4), we next asked whether inhibition of GR signaling blocks the sensitized LH^Penk neuronal reactivity of PSS mice upon HFD exposure. After injecting AAV-DIO-eGFP into the LH of Penk-Cre mice, we implanted a cannula into the same site for the local infusion of mifepristone (Fig. 8i). We found that up to 25.42% of LH^Penk neurons show HFD-induced c-fos expression in vehicle-treated PSS mice, but a local infusion of mifepristone reduced the induction of c-fos, resulting in only 5.78% of LH^Penk neurons showing HFD-induced c-fos expression (Fig. 8j, k). These data support the hypothesis that PSS-driven activation of GR signaling in the LH is required for sensitizing LH^Penk neuronal activity to HFD, leading to the emotional overeating toward palatable food following PSS exposure.

## Discussion

Perceptions of threats trigger both instant and delayed stress responses. For example, threats trigger immediate emotional reactions such as a strong feeling of fear or aversion that help the individuals actively avoid the source of the threat[47,48]. Threats can often trigger maladaptive responses that persist well after the threatening situation is over, including reactions like cravings for palatable foods or pathological eating habits[49,50]. Thus, identifying the post-stress factors that increase vulnerability to the overconsumption is an important step for developing therapeutics to combat problematic eating habits, such as binge eating or bulimic episodes.

Coping with stressful situations can boost hedonic hunger triggered by psychological needs, instead of actual needs from physical hunger, resulting in increasing motivation to consume palatable foods[51]. Here, we have identified LH^Penk neurons as a critical component for eliciting emotional overeating in mice exposed to a naturally threatening stimulus like PSS. The tendency to develop overeating was particularly exacerbated under HFD exposure, as demonstrated by our data showing that PSS or chemogenetic activation of LH^Penk neurons increased HFD consumption but had no effects on *ad libitum* NC intake (Figs. 1, 4). Our discovery of LH^Penk neuron-mediated overeating targeted selectively to palatable food presents a previously unknown mechanism by which the threat-induced behavioral stress responses influence the hypothalamic neuropeptidergic systems that control emotionally triggered eating, distinct from homeostatic eating.

In the LH, Penk-expressing neurons are mostly distinct from the Pmch-, Hcrt- or Lepr-expressing neuronal populations (Fig. 2), suggesting that LH^Penk neurons may play unique roles in regulating feeding behaviors regulated by the LH. Notably, our findings reveal that the LH enkephalin is the critical endogenous opioid source for mediating PSS-induced maladaptive responses at the neuronal and behavioral level (Fig. 5). Given previous studies demonstrating that the enkephalin particularly acts as a direct or indirect modulator for transmission of multiple neurotransmitter systems[52], we speculate that the LH enkephalin possibly acts on GABAergic or glutamatergic inputs through retrograde transmissions, thereby enhancing LH^Penk neuronal reactivity to HFD eating after PSS exposure. In addition, we found that LH^Penk neurons receive major glutamatergic inputs from the PVN (Supplementary Fig. 2). Because the PVN is a primary driver of the HPA axis, which regulates CORT levels in response to stressful events, engaging PVN-LH^Penk neuronal circuit activity may override a predisposition to PSS-triggered HFD overconsumption. We also found that the PVN is one of the hypothalamic subregions that showed enhanced c-fos induction in response to HFD after PSS exposure (Supplementary Fig. 1). This raises the possibility that the PSS-primed PVN can be a critical upstream structure that sensitizes LH^Penk neuronal reactivity to HFD. Future extended interrogation into whether LH endogenous enkephalin influences PVN glutamatergic inputs will be required.

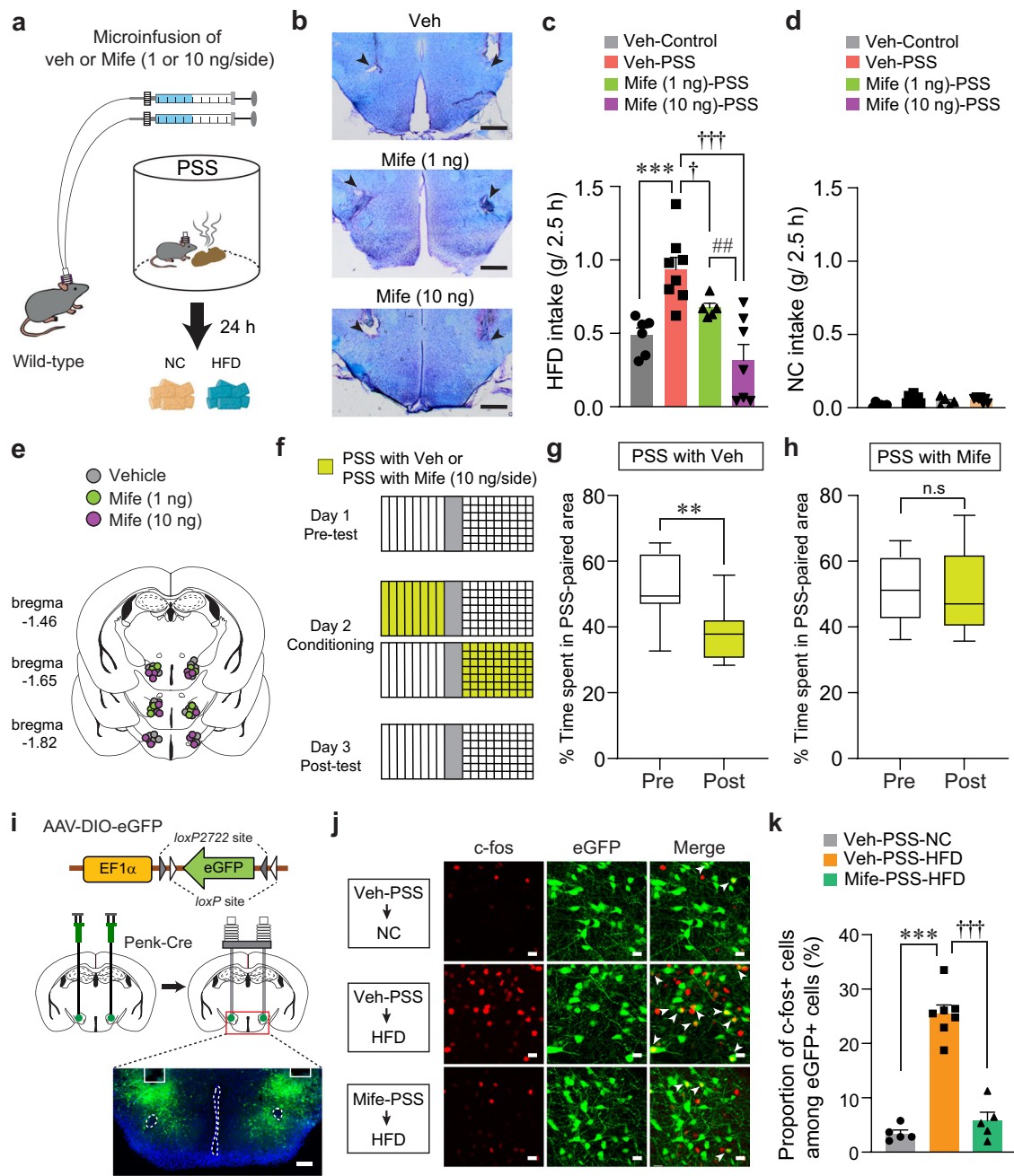

**Fig. 8 | Inhibition of GR signaling in the LH prevents PSS-induced maladaptive behaviors. a** Schematics for drug microinfusions before PSS exposure. **b** Images show locations of cannula tips in the LH. Arrowheads denote the end of tips, replicated independently with similar results in 4 mice. Scale bars, 500 μm. **c, d** 2.5 h food consumption 24 h after exposure to either control or PSS ($n = 6, 8, 5$, and 7 mice). In (**c**), one-way ANOVA ($F_{(3,22)} = 11.778$, $p < 0.001$) was followed by Fisher LSD post hoc test for multiple comparisons; ***$p < 0.001$ compared with veh-injected control mice; †$p = 0.041$, †††$p < 0.001$ compared with veh-injected PSS mice; ##$p = 0.008$ compared with Mife (10 ng/side)-injected PSS mice. **e** Locations of the injection cannula tips in the mice included in (**c, d**). **f–h** Experimental procedure to test CPA (**f**). Time spent (%) of veh- (**g**; $n = 8$ mice) and Mife-treated mice (**h**; $n = 6$ mice) in PSS-paired chambers during pre- and post-test. Box–whisker plots display median (center) and 2.5 to 97.5 percentiles of the distribution (bounds) with

whiskers extending from min to max values. In (**g**), two-tailed paired $t$ test, $t_7 = 4.380$, **$p = 0.00324$. In (**h**), two-tailed paired $t$ test, $t_5 = 0.476$, $p = 0.654$. **i** Schematic for the bilateral injection of AAV-DIO-eGFP, followed by implantation of cannula into the LH of Penk-Cre mice, replicated independently with similar results in 5 mice. Scale bar, 200 μm. **j** Representative images showing c-fos immunoreactivity and eGFP in the LH in response to either NC or HFD 24 h after PSS exposure. White arrowheads indicate the colocalization of c-fos with Penk expression, replicated independently with similar results in 5 mice. Scale bars, 20 μm. **k** Quantification of the proportion of c-fos-positive cells among LH$^{Penk}$ neurons in (**j**) ($n = 5, 7$, and 5 mice per group). One-way ANOVA ($F_{(2,14)} = 71.860$, $p < 0.001$) was followed by Fisher LSD post hoc test for multiple comparisons; ***$p < 0.001$, †††$p < 0.001$ compared with veh-treated PSS mice after HFD exposure. n.s not significant. Data are expressed as mean ± SEM.

We identified the MORs in the LPAG as possible targets that relay information from LH[Penk] neurons and causally mediate PSS-induced HFD overconsumption (Fig. 6). However, PSS mice given the pharmacological inhibition of MORs in the LPAG still exhibited avoidance behavior for PSS-paired compartment in CPA procedure, suggesting that alternative downstream targets of LH[Penk] neurons may exist to drive PSS-induced negative emotional state. For example, we showed that LH[Penk] neurons make dense efferent connections with the VTA (Fig. 6), a brain region that mediates motivated behaviors by signaling positive or negative motivational stimuli[36]. Given that the LH is thought to be a key afferent input providing information about aversive outcomes to the VTA[53], it is plausible that enkephalin signaling on the LH[Penk] – VTA projection may play a role in controlling negative emotional valence following PSS exposure. Future studies focusing on endogenous enkephalinergic system on the projection-specific LH[Penk] circuit organizations may account for its possible discrete effects on various aspects of disordered eating behaviors.

Numerous studies showed that neurons that increase food consumption are able to transmit positive valence in the absence of food[54,55]. Influential experiments with brain stimulation identified the LH neurons that elicit food intake and encode positive valence[9]. Molecularly defined CeA neuronal circuits upregulate feeding and reward, promoting positive reinforcement[34,56]. In contrast to most of the studies that have addressed food consumption associated with positive emotional valence, fewer studies have suggested a role for neuronal systems in driving food-seeking behaviors associated with negative emotional valence[57]. However, it is important to note that exposure to life-threatening events can generate aversive distress states that motivate palatable food overconsumption[58,59]. For example, binging behavior[60], such as eating an excessive amount of food until the point of discomfort or pain, can be initiated by unpleasant or overwhelming emotions. We found in mice that LH[Penk] neuronal activation encodes negative emotional valence and elicits copious HFD consumption, while silencing of the same neurons suppresses PSS-induced HFD overconsumption accompanied by aversive emotional states (Fig. 4). This supports the hypothesis that the primary function of LH[Penk] neurons in the pathophysiology of stress-induced maladaptive eating behaviors lies in mediating the aversive motivation that drives palatable food consumption.

The HPA axis -the principal neuroendocrine mechanism by which the body responds to stressful events- is crucial in the time-dependent regulation of appetite after stress. In the first minutes after a stressful event, CRH is released at the apex of the HPA axis, which then acutely discourages food-seeking because escape or defense behaviors are more pressing[61,62]. On the contrary, glucocorticoid hormones, the end product of HPA axis, exert the opposite effect via stimulating appetite or increasing comfort eating hours to days after a stressful event[63,64]. Our data indeed demonstrated that CORT-driven HFD overconsumption is clearer a day after, but not immediately after the CORT administration (Supplementary Fig. 7).

Likewise, we found that CORT increases LH[Penk] neuronal reactivity to HFD 24 h after the drug treatment. Previous studies suggest that the CORT and its receptor GR signaling have the ability to change synaptic plasticity through modulating α-amino-3-hydroxy-5-methyl-4-isoxazolepropionic acid receptor (AMPAR)-mediated synaptic transmissions via two mechanisms: 'delayed' genomic processes involving transcriptional regulation of GR-affected target genes and 'rapid' non-genomic processes through activation of kinase signaling pathways[65,66]. Given our data showing that CORT pretreatment elicited delayed onset of emotional eating accompanied by increased responsiveness of the LH[Penk] neurons (Fig. 7), it is plausible that CORT may increase the surface expression of AMPAR subunits in the LH[Penk] neurons possibly via delayed genomic mechanisms. This process may include increased transcriptional regulation of genes functionally involved in GluR2/3 delivery and/or membrane anchoring.

Investigations into the CORT-mediated genomic processes in the LH and multiple candidate genes that could influence AMPA receptor trafficking in the LH[Penk] neurons will further our understanding of cellular/synaptic mechanisms underlying delayed-onset emotional overeating after PSS exposure.

In summary, our findings explain how LH[Penk] neurons can drive emotionally triggered HFD overconsumption after PSS exposure. We propose that interactions between elevated CORT signaling and endogenous enkephalinergic LH circuits mediate the effects of PSS on emotional overeating. LH endogenous enkephalin peptides are critical for sensitizing LH[Penk] neuronal reactivity to palatable foods, leading to HFD overconsumption associated with an aversive emotional state. Conversely, when the LH[Penk] neuronal activity is silenced, the negative emotional state that normally follows PSS is attenuated, resulting in reduced emotional eating. Thus, our findings suggest that suppression of endogenous enkephalinergic LH circuit activity or GR antagonism may be effective interventions for the emotional overeating that often follows life-threatening events.

## Methods

### Mice

All experiments were conducted with 12–20-week-old wild-type C57BL/6J, Penk-Cre (stock no. 025112), and Lepr-Cre (stock no. 008320) mice, which were attained from the Jackson Laboratory. Penk-Cre and Lepr-Cre mice for all experiments were crossed with C57BL/6J mice for several generations before use. To visualize the Penk-positive neurons, Penk-Cre mice were crossed with Ai14 mice (stock no. 007908; tdTomato reporter line; Jackson Laboratory). Male and female mice were used for all experiments, including immunohistochemistry, FISH, anatomical tracing, food intake measurements, and behavioral experiments. Mice were group housed on a 12-h light/dark cycle with standard bedding in a temperature- and humidity-controlled room (-21 °C and 42% humidity). All behavioral experimentations were performed during the light cycle. Experimenters were blinded to the group allocation and outcome assessment. All experiments were approved by the Institutional Animal Care and Use Committee (IACUC) of the Virginia Polytechnic Institute and State University.

### Predator scent stimulus (PSS) exposure

The PSS exposure was adopted from previously published methods[27] with minor modifications. Briefly, mice received a single exposure to PSS in a covered, clear cylindrical Plexiglas chamber (24 cm diameter, 24 cm height). Prior to each PSS session, a small wire mesh cage containing well-soiled cat litter was placed in the center of the chamber. Mice were individually placed in the test chamber for 10 min. The control mice were exposed to a small wire mesh cage containing unused new litter for the same amount of time. The test chambers were separately used for control mice and cleaned with 70% ethanol between sessions. A video tracking software (ANY-maze) was used to monitor the movement of mice in the test chambers, which were subdivided into two zones, a center (16 cm diameter circle) and a periphery.

### Food intake measurements

Mice were group-housed (2–5 per cage) and standard NC (18% kcal% fat; no. 2918, Harlan-Teklad) was always available *ad libitum* before the experiment. To assess food intake levels, mice were singly housed either immediately or 24 h after PSS exposure, subsequently followed by exposure to both NC and HFD (60% kcal% fat; D12492, Research Diets) *ad libitum*. Mice then had free-choice access to both NC and HFD. Food intake levels were measured 2.5 h later by subtracting the amount of food remaining in cages from the weight of foods initially provided. For chemogenetic activation experiments, mice were individually housed and given free-choice access to the NC and HFD

immediately after intraperitoneal injections of saline or CNO (2 mg/kg). Food intake measurements were taken 2.5 h after the drug administration.

### Conditioned place avoidance (CPA)

A three-chamber rectangular apparatus ($40.5 \times 21 \times 35$ cm$^3$) containing two chambers ($18 \times 21 \times 35$ cm$^3$) on opposite sides and a central chamber ($4.5 \times 21 \times 35$ cm$^3$) was used. The two chambers displaying different wall patterns (stripes versus dots) were segregated by two removable vertical inserts with the same pattern on each side. On day 1 (pre-test), the mouse was placed in the central chamber and allowed to freely explore the open arena, including all chambers for 20 min. Based on these pre-test results, mice with >65% side preference were excluded from further study. On day 2, for the 60-min conditioning session, the mouse was confined to one side of the chamber where the PSS-containing small wire mesh cage was located. The PSS-paired side was chosen randomly and counterbalanced across testing. On day 3 (post-test), the mouse freely explored the open arena again for 20 min. The amount of time spent in each chamber and the total distance traveled were assessed during the pre- and post-test using a camera (Logitech) and video tracking software (ANY-maze). The percentage of time spent in the PSS-paired chamber was calculated as [(time in the PSS-paired chamber)/(time in the PSS-paired chamber + time in the unpaired chamber) × 100 (%)]. For CPA experiments involving chemogenetic activation, the behavioral procedures were identical, but on days 2–5, morning and afternoon sessions were carried out. During the morning session, the mouse was injected with vehicle (sterile distilled water, i.p) and confined to one designated chamber for 30 min. In the afternoon, the mouse was injected with CNO (2 mg/kg, i.p) and confined to the opposite chamber for 30 min. The percentage of time spent in the CNO-paired chamber on day 1 (pre-test) and day 6 (post-test) was calculated as [(time in the CNO-paired chamber)/(time in the CNO-paired chamber + time in the vehicle-paired chamber) × 100 (%)].

### Locomotion

Locomotion was assessed in an open field arena ($40.5 \times 40.5 \times 35$ cm$^3$). Mice were placed individually in the arena and allowed to move around freely for 30 min. Locomotor activity was monitored using a camera (Logitech) positioned above the arena and analyzed by video tracking software (ANY-maze).

### Buried food olfactory test

This experiment was performed as described in previous studies[67] with minor modifications. Briefly, mice were given strawberry cookies and water ad libitum for 2 days before testing. Mice were deprived of cookies for 12 h before testing. During the test, a new cage was prepared with clean bedding (3 cm high), and a small piece of the cookie was buried at the corner of the cage under 2 cm of bedding. The location of the cookie was changed randomly on each test session. The animal was introduced to the cage at the opposite end of the cookie burial site. The time for the mouse to find the buried cookie and begin to burrow was measured. Fresh cages and bedding were used for every test.

### Optogenetic stimulation and behavioral tests

For ChR2-mediated activation of LH$^{Penk}$ neurons, a 3-m-long fiber-optic patch cord (200 μm, 0.57 NA, Doric Lenses) was connected to the chronically implanted optic fiber and suspended above the behavioral testing area to allow animals to move freely while receiving 473-nm blue light stimulations (Doric Lenses). To perform a real-time place test (RTPT), mice were placed in a three-chamber rectangular apparatus ($40.5 \times 21 \times 35$ cm$^3$), which had a central chamber and two side chambers ($18 \times 21 \times 35$ cm$^3$) distinguished by wall patterns (stripes versus dots). On a baseline day, mice explored the entire arena for 20 min without optical stimulation (no light). Most mice displayed no

preference, and only those with >65% side preference on the baseline test were excluded from further study. On a subsequent day, the mice were placed into the no-stimulation side of the arena, which was randomly assigned. The 20-min sessions began when the mouse first fully crossed (all four paws) from the 'no-stimulation' side to the 'stimulation-paired side.' Upon crossing to the stimulation-paired side, mice received stimulation (blue light in a high-frequency train; 10 ms pulses of 20 Hz; 10–15 mW at tips) until returning to the no-stimulation side, upon which stimulation ceased. The time spent in each chamber and the total distance traveled were assessed using a camera (Logitech) and video tracking software (ANY-maze). The percentage of time spent in the stimulation-side chamber was calculated as [(time in the stimulation-paired chamber)/(time in the stimulation-paired chamber + time in the no stimulation-paired chamber) × 100 (%)].

For locomotion tests with optogenetic stimulations, the mice were individually placed in an open field arena ($40.5 \times 40.5 \times 35$ cm$^3$) and the total distance traveled was measured for 15 min (ANY-maze).

For ChR2-mediated food intake measurement, NC was always available ad libitum before the testing. During a 30-min test session, mice were individually placed in their homecage containing both NC and HFD, and received 10 min each of pre-photostimulation (light-off), photostimulation (light-on; blue light in a high-frequency train; 10 ms pulses of 20 Hz; 10–15 mW at tips), and post-photostimulation (light-off). The initial and final weight of the food were recorded to determine the total amount of food consumed (g) after each 10 min session.

### Serum corticosterone measurement

Mice were anesthetized with isoflurane and decapitated either 0.5 h or 24 h after PSS exposure. Whole trunk blood samples were collected in 1.5 ml microcentrifuge tubes and allowed to clot at room temperature for 30 min. Tubes were then spun for 15 min at 3000 g at 4 °C. Supernatant containing clear serum was stored at −80 °C until an enzyme-linked immunosorbent assay was applied to measure serum corticosterone using an assay kit (Invitrogen, EIACORT).

### Immunohistochemistry

Mice were individually housed, and the HFD was introduced to the mice for 2.5 h. For c-fos immunoreactivity, mice were subsequently anesthetized with isoflurane and perfused with 4% paraformaldehyde (PFA). Brains were obtained and post-fixed overnight in 4% PFA. The brains were sectioned coronally (50 μm) using a vibratome, and the sections were washed in PBS containing 0.3% Triton X-100 (PBS-T, pH 7.4) and incubated in 1% BSA (vol/vol) in PBS-T for 1 h. The sections were then immunostained using a rabbit anti-c-fos antibody (1:5000 dilution; Cell Signaling Technology) applied overnight in PBS-T at room temperature. The next day, the sections were washed in PBS-T and incubated in a 1:1000 dilution of Alexa Fluor Plus 555 anti-rabbit secondary antibody (Thermo Fisher Scientific) in PBS-T for 1 h. Subsequently, sections were rinsed with PBS-T and mounted using a mounting medium with DAPI. Images were acquired using a Zeiss LSM700 confocal microscope (Carl Zeiss) and quantitatively analyzed with ImageJ. For c-fos quantification in the LH, sections were obtained from Bregma −1.34 to −1.82 mm from each animal. The neurons located lateral and up to 0.1 mm medial to the fornix or up to 0.2 mm above or below the fornix were considered to be in the LH. Cell counts were made within this reference area by applying equal thresholds to all images and using the 'analyze particles' function in ImageJ.

### RNA in situ hybridization

Mice were deeply anesthetized using isofluorane inhalation and perfused with ice-cold PBS. Brains were rapidly extracted and flash-frozen with isopentane (Sigma-Aldrich) pre-chilled with dry ice in 70% ethanol. Coronal brain slices (18 μm) containing the LH or PVN were sectioned using a cryostat (Leica CM1860) at −20 °C. Brain slices were placed directly onto slides and stored at −80 °C until RNA in situ

hybridization, which was conducted using RNAscope probes (Advanced Cell Diagnostics, ACD). Slides were fixed in 4% PFA for 15 min at 4 °C and subsequently dehydrated for 5 min with 50%, 70%, and 100% ethanol at room temperature. Sections were then incubated with a Protease IV solution for 30 min and washed with 1× PBS before being incubated with probes for 2 h at 40 °C in the HybEZ oven (ACD). All probes used were commercially available: Mm-Penk (318761), Mm-Pmch (478721), Mm-Hcrt (490461), Mm-Lepr (402731), Mm-Slc32a1 (VGAT; 319191), Mm-Slc17a6 (vGlut2; 319171), Cre (312281), EGFP (400281), Mm-Nr3c1 (GR; 475261), Mm-Nr3c2 (MR; 456331). After washing twice with wash buffer, the signals of the tissue sections were amplified in the amplification buffers at 40 °C. After the final rinse, DAPI solution was applied to the sections. Slides were visualized with a Nikon SoRa inverted spinning disk confocal microscope.

## Reverse transcription and quantitative PCR (qPCR)

Mice were anesthetized with isofluorane and 250 µm slices were prepared in PBS, using a vibratome (Leica VT1200). The LH was microdissected bilaterally, and samples were immediately frozen on dry ice and stored at −80 °C prior to RNA isolation. Total RNA was extracted from dissected samples using a Hybrid-R RNA purification kit (GeneAll Biotechnology). Purified RNA samples were reverse transcribed by using the SuperScript-IV First-strand cDNA synthesis kit (Invitrogen). qPCR was performed by using TaqMan Gene Expression Assay Kit (Applied Biosystems). All TaqMan probes were purchased from Applied Biosystems and are as follows: Penk (Mm01212875_m1), Lepr (Mm00440181_m1), Hcrt (Mm01964030_s1), Pmch (Mm01242886_g1), GR (Nr3c1; Mm00433832_m1) and glyceraldhyde-3-phosphate dehydrogenase (GAPDH; Mm99999915_g1). Target amplification was performed by using QuantStudio 3 Real-Time PCR System (Applied Biosystems) with QuantStudio Real-Time PCR software. The relative mRNA expression levels were calculated via a comparative threshold cycle (Ct) method using GAPDH as an internal control: $\Delta Ct = Ct$ (gene of interest) − Ct (GAPDH). The gene expression fold change, normalized to the GAPDH and relative to the control sample, was calculated as $2^{-\Delta\Delta Ct}$.

## Plasmid and virus generation

AAV vector plasmids were generated using standard molecular cloning methods or were obtained from external sources. AAV viruses used in this study were produced in-lab, as previously described[68]. Briefly, AAV was produced by transfection of 293 cells with three plasmids: an AAV vector expressing target constructs including AAV-EF1α-DIO-eGFP, AAV-EF1α-DIO-GCaMP6, AAV-EF1α-DIO-mRuby2-TVA, AAV-EF1α-DIO-RVG, AAV-hsyn-DIO-mCherry, AAV-hsyn-DIO-hM3D(Gq)-mCherry, AAV-CAG-DIO-eGFP-Kir2.1, AAV-EF1α-DIO-EmGFP-Penk shRNA, AAV-CMB-FLEX-SaCas9-U6-sgPenk (Addgene plasmid #159916), AAV-EF1α-DIO-eYFP, AAV-EF1α-DIO-ChR2-eYFP, AAV-hsyn-DIO-GR), AAV helper plasmid (pHELPER; Agilent), and AAV rep-cap helper plasmid (pRC-DJ). At 72 h after transfection, the cells were harvested and lysed. Next, the viral particles were purified by an iodixanol step-gradient ultracentrifugation method. The iodixanol was diluted, and the AAV was concentrated using a 100-kDa molecular mass cutoff ultrafiltration device. The genomic titer was validated by qPCR. The AAV vectors were diluted in 1 × PBS to a working concentration of approximately $10^{12}$ viral particles/ml. EnvA-pseudotyped glycoprotein (G)-deleted rabies virus expressing eGFP (RVΔG-eGFP) was purchased from the Gene Transfer, Targeting, and Therapeutics Facility at the Salk Institute for Biological Studies.

To construct shRNA against Penk (NM 001002927.3), oligonucleotides that contained 21 base-pair sense and antisense sequences (5′-CTCCTCCGATCTGCTGAAAGA-3′) targeting Penk were connected with a hairpin loop followed by a poly(T) termination signal. This shRNA oligonucleotide was ligated into BLOCK-iTTM POLII miR RNAi expression vectors (Invitrogen) and then transferred to an AAV vector together with EmGFP. For testing the efficacy of the shRNA, we stereotaxically injected AAVs expressing Penk shRNA into the LH. Two weeks after the injection, the virus-infected area was dissected and qPCR was performed to measure Penk mRNA levels (Fig. 5b).

To overexpress GR, AAV vectors carrying GR (Nr3c1; NM 008173.4) were cloned. A mouse cDNA was amplified by PCR using the following primer sets: GR-F: 5′-ATGGACTCCAAAGAATCCTTAG-3′ and GR-R: 5′-TCATTTCTGATGAAACAGAAG-3′; The amplified fragment was inserted into NheI and AscI restriction enzyme sites of the AAV vector to produce AAV-hsyn-DIO-GR.

## Stereotaxic surgeries and histology

Penk-Cre, Lepr-Cre or wild-type mice were anesthetized with a combination of ketamine (100 mg/kg) and dexmedetomidine (0.5 mg/kg). The mice were mounted in a stereotaxic frame (David Kopf Instruments). Body temperature was kept stable using a heating pad (33 °C) while recovering from anesthesia. Viral injection targets were determined using coordinates based on the Paxinos and Franklin mouse brain atlas[69].

For in vivo Ca²⁺ imaging experiments, Penk-Cre animals received an unilateral injection of AAV-DIO-GCaMP6f (300 nl) into the LH (AP −1.65, ML ± 1.12, DV −5.25) of Penk-Cre mice. A sterile 26-gauge needle was gradually lowered into the same location to a depth of −5.13 mm from the cortical surface to make a path for GRIN lens implantation. The virus was then delivered using a Hamilton microsyringe connected to borosilicate glass pipettes at a rate of 100 nl/min controlled by a syringe pump (PHD Ultra, Harvard Apparatus). Subsequently, the ProView Integrated GRIN lens (diameter 0.5 mm, length 8.4 mm, Inscopix) was implanted 80–100 µm above the virus injection site. The implanted GRIN lens and microendoscope baseplate were secured to the skull with an initial layer of adhesive cement (C&B metabond; Parkell) followed by a second layer of dental cement (Ortho-Jet; Lang). Sterile tissue adhesive (Vetbond; 3 M) was used to close the incision. A baseplate cover (Inscopix) was used to protect the lens when not in use. In vivo Ca²⁺ imaging tests were performed 6 weeks after the ProView Integrated GRIN lens implantation.

For input mapping experiments with EnvA-pseudotyped rabies virus, 1:1 mixture of AAV-DIO-mRuby2-TVA and AAV-DIO-RVG was unilaterally infused into the LH of Penk-Cre mice. After allowing 2 weeks for viral expression, animals were again anesthetized as previously described, and EnvA-pseudotyped RVΔG-eGFP (250 nl) was infused into the same site of AAVs injections. Six days post EnvA-pseudotyped RVΔG-eGFP infusion, animals were sacrificed for input mapping analysis. For anatomical output mapping, AAV-DIO-eGFP (300 nl) was unilaterally injected into the LH of Penk-Cre mice. Mice were sacrificed 2 weeks after the injection for output mapping analysis.

For the manipulations of LH^Penk neuronal activity for behavioral testing, Penk-Cre animals were bilaterally injected with 350 nl of AAV-DIO-mCherry, AAV-DIO-hM3D-mCherry, AAV-DIO-Kir2.1-eGFP, or AAV-DIO-eGFP into the LH. Two weeks were given before behavioral tests. For optogenetic behavioral experiments, Penk-Cre animals were bilaterally injected with 350 nl of AAV-DIO-eYFP or AAV-DIO-ChR2-eYFP into the LH. Bilateral optic fibers (200 µm diameter, 0.66 NA; Doric Lenses) were implanted into the LH. Implanted fibers were fixed to the skull with adhesive and dental cement, as described above. Two weeks were given for viral expression in cell bodies before behavioral tests. Viral injections and fiber placements were confirmed after the behavioral tests were completed.

For shRNA-related testings, viral preparations (AAV-DIO-EmGFP, AAV-DIO-EmGFP-Penk shRNA) in a 300–350 nl volume were injected bilaterally into the LH of Penk-Cre mice at a slow rate (100 nl/min) using a syringe pump. Mice were allowed 2 weeks to recover from the virus injections before starting of either behavioral tests or immunostaining experiments. Injection sites were confirmed in all animals by preparing coronal sections containing

the desired plane, and animals with incorrect injection placement were excluded from analyzes.

For CRISPR/SaCas9 viral-based system, AAV carrying a sgRNA targeted to the mouse Penk locus along with SaCas9 (AAV-CMV-FLEX-SaCas9-U6-sgPenk) was bilaterally injected into the LH of Penk-Cre mice. Two weeks were given for viral expression before behavioral tests.

For microinjections of the mifepristone and CTAP, a guide cannula (26-gauge, 6-mm long; Protech International Inc.) was chronically implanted into the LH and LPAG (AP −4.45, ML ± 0.50, DV −1.9), respectively. Implanted cannulae were secured to the skull as described above, and then obturators were placed in the guide cannulae. Behavioral testings were performed 2 weeks after the implantation. Upon completion of all behavioral experiments, cannula placements were confirmed. Mice with off-target implant tip locations were excluded from the final analyzes.

After each experiment, the extent of viral transduction spread was validated to determine whether it was restricted to the LH without off-target effects in other adjacent areas, such as the Arc, VMH, and DMH. The inclusion standards were applied when the virally-labeled neurons were located in the LH reference area, that is, lateral and up to 0.1 mm medial to the fornix or up to 0.2 mm above or below the fornix. If the viral expression was found outside this reference area or the viral transduction was weak in the LH (covering less than 50% of the total LH area), the mice were excluded from the final dataset, which was determined by two experimenters blinded to the experimental design.

### In vivo Ca²⁺ imaging and behavior in freely moving mice

For in vivo $Ca^{2+}$ imaging experiments, mice were habituated in a clear cylindrical Plexiglas chamber (20 cm diameter, 25 cm height) for 30 min with the head-mounted Inscopix miniature microscope (nVoke 2.0) attached to a commutator to prevent the cord from tangling. Prior to data acquisition, scope focus, EX-LED power, and gain were optimized for each mouse. These settings remained constant throughout the experiment. To monitor the GCaMP6f fluorescence change in freely moving animals, physical stimuli (HFD, Lego brick, new litter- or PSS-containing small wire mesh cage) were each presented in the chamber with a 2–5 min recording of baseline fluorescence. Mice were allowed to freely investigate the physical stimuli, during which GCaMP6f fluorescence was recorded. All mice behaviors were recorded by a camera (Logitech) concurrently. To test whether the in vivo activity of individual LH^Penk neurons was altered by acute CORT injection, the numbers of $Ca^{2+}$ transients per min were measured during 4-min periods after CORT injection (at 5, 15 min post-injection) and compared to the numbers of $Ca^{2+}$ transients per min during a 4-min pre-injection period.

### In vivo Ca²⁺ imaging data processing and analysis

Inscopix Data acquisition software (IDAS, version 1.6.1) was used to acquire TIFF images of fluorescence dynamics at 20 frames per second. The EX-LED power was maintained at 0.9–1.4 mW/mm² with a gain of 3.5. Inscopix Data processing software (IDPS, v 1.6.1) was used to pre-process $Ca^{2+}$ data from each imaging session[70]. Briefly, $Ca^{2+}$ imaging data were down-sampled temporally (2 × temporal bin) and spatially (4 × spatial bin) and motion-corrected for rigid brain movement. In maximum projection images, regions of interest (ROI) corresponding to cell bodies were identified using an automated cell-finding function and were visually inspected to ensure accuracy. Raw fluorescence traces for individual LH^Penk neurons were then z-normalized using the mean and standard deviation of the entire imaging session. The normalization of individual neuronal $Ca^{2+}$ dynamics was performed using Inscopix Python (v3.8) scripts or customized MATLAB-based software (Mathworks) for further analysis.

For the classification of neurons in HFD and novel object experiments, the basal z-score level and standard deviation (σ) were determined for 10 s before the 1st behavioral onsets (for example, HFD eating bout or physical contact with a novel object). Cells were considered as activated or inhibited if the mean z-score during the 10 s after the 1st behavioral onset was higher than [basal z-score + 0.5 σ] or lower than [basal z-score − 0.5 σ], respectively. Cells with mean z-score between [basal z-score + 0.5 σ] and [basal z-score − 0.5 σ] were categorized as unresponsive. For the classification of neurons in PSS and empty container experiments, all the analysis processes were identical, but the mean z-score level and standard deviation (σ) were determined for 1 min before and after the introduction or removal of PSS or an empty container.

To detect individual $Ca^{2+}$ transients[68,71], we processed the time series of individual neuronal fluorescence traces to remove slow drifts, estimated by a median filter of 10 s in width. Next, the median absolute deviation (MAD) of the entire time series was computed. $Ca^{2+}$ transients were extracted by sequentially detecting each upward transient that exceeded a 5.5-MAD threshold (equivalent to 4 σ assuming normal distribution) and following the previous event by an interval no shorter than 2 s.

Discrete timestamps for the onset of HFD eating or novel object physical contact were manually scored from simultaneously recorded behavioral videos by two experimenters blinded to the experimental design. For correlating LH^Penk neuronal activity with eating bouts, $Ca^{2+}$ transients of cells from individual animals aligned to the 1st eating bout onset. During 1.5 min after the 1st eating bout, the numbers of $Ca^{2+}$ transients that occurred during eating bouts were counted. The percentage of the correlated or non-correlated $Ca^{2+}$ transients was calculated among the total numbers of $Ca^{2+}$ transients generated from LH^Penk neurons during the 1.5 min after the 1st eating bout.

### Drug treatment and histology

For chemogenetic manipulations, CNO (Enzo life science) was dissolved and diluted in sterile distilled water. Mice were administered CNO (2 mg/kg, i.p) or an equivalent volume of saline immediately before subjected to HFD. For pharmacological manipulation, CORT (2 mg/kg; Sigma-Aldrich) and mifepristone (1 or 10 ng/0.5 μl/side; Tocris Bioscience) were dissolved in distilled water containing 0.8% dimethyl sulfoxide (DMSO) and 7.2% Tween-80. Mice were subcutaneously injected with CORT or vehicle and subjected to HFD either immediately or 24 h after the drug treatment. For intracranial infusion of mifepristone, an internal cannula (33-gauge, projecting 0.2 mm below the tip of the guide; Protech International Inc.) was connected to 1 mL Hamilton syringes via polyethylene (PE)−20 tube. Vehicle or mifepristone was microinjected bilaterally into the LH 5 min before CORT administration (2 mg/kg, s.c). For experiments involving PSS exposure, mifepristone was microinfused 30 min before subjecting mice to a PSS-containing chamber. To assess the effects of drug treatment on locomotor activity, a microinfusion of mifepristone was performed 24 h before locomotion tests.

CTAP (0.1 or 1 μg/0.5 μl/side; Tocris Bioscience) was dissolved in distilled water. Vehicle or CTAP was microinjected bilaterally into the LPAG 5 min before CNO administration (2 mg/kg, i.p). For experiments involving PSS, CTAP was microinfused 30 min before subjecting mice to HFD 24 h after PSS exposure. For locomotion or CPA experiments, CTAP was microinfused 30 min before subjecting mice to either an open field or a CPA chamber, respectively.

Upon completion of experiments, mice were anesthetized and perfused with 4% PFA. Brains were collected and post-fixed in 4% PFA overnight. Coronal sections (50 μm) were subsequently stained with 0.2% cresyl violet (Sigma-Aldrich) for the verification of cannula tip locations. Only mice with cannula tips located bilaterally in the LH or LPAG were included in the data analysis.

## Statistics

Animals used in this study were not selected based on prerequisite features other than general animal well-being (for example, regular grooming and locomotion, no apparent infections) for allocation into a particular experimental group. Differences across more than two groups were analyzed with one-way or two-way ANOVA followed by Bonferroni or Fisher's LSD post hoc tests for multiple comparisons. For comparisons between two groups, two-tailed t-tests (paired or unpaired) were used, as described in the figure legends. The distribution of data was checked for normality and equal variance. $P < 0.05$ was considered statistically significant. Data are reported as the mean ± s.e.m. All statistical tests were performed using SigmaStat software (version 4.0, Inpixon).

## Reporting summary

Further information on research design is available in the Nature Portfolio Reporting Summary linked to this article.

## Data availability

The processed data underlying the figures are provided with this paper as Source data file. Source data are provided with this paper.

## Code availability

All custom codes used for in vivo calcium imaging analysis in this manuscript are available on Github (https://github.com/Shinlab/matlabcodes).

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

## Acknowledgements

We acknowledge Dr. Liu for consulting in vivo calcium imaging data analysis. We also thank N. Barakat, M. Milam and M. Cawley for technical assistance with analyzing behavioral, anatomical, and molecular data. Food illustrations were created with BioRender.com. This study was supported by grants from National Institutes of Health (NIH) R01DK132566 to S.S. Fralin Biomedical Research Institute Seale Innovation fund awards to S.S. and the integrated Translational Health Research Institute of Virginia (iTHRIV) funding to S.S. in part by the National Center for Advancing Translational Sciences, NIH, through grant KL2TR003016/UL1TR003015.

## Author contributions

I-J.Y., Y.B. and S.S. designed the study and interpreted the results, and prepared the manuscript. I-J.Y. performed the behavioral, in vivo calcium, pharmacological experiments and analyzed the data in consultation with S.S; Y.B. conducted testing for behavioral/molecular manipulations, FISH experiments and analyzed the data in consultation with S.S; A.R.B. performed technical assistance.

## Competing interests

The authors declare no competing interests.
