## [Peer Review File · Nature Communications]

Lateral hypothalamic proenkephalin neurons drive threat-induced overeating associated with a negative emotional stateREVIEWER COMMENTS

Reviewer #1 (Remarks to the Author):

In this study, You et al. demonstrated LH Penk cells as a critical population for controlling stress-induced overeating. They showed that predator cues induced acute anxiety, conditioned avoidance and overeating of palatable food 24 hours later. The authors then demonstrated LH Penk cells can bi-directionally modulate the HFD overeating and conditioned avoidance. In vivo calcium recording further demonstrated an increased activity of LH Penk cells to HFD after predator odor exposure. The authors went one step further to demonstrate that Cort is an important mediator for the predator odor induced HFD overeating. Blocking GR in LH can block Cort or predator odor induced HFD overeating. The study is well performed and novel. However, the reviewer was left uncertain whether LH Penk cells are indeed unique in mediating the predator odor-induced overeating or Penk simply marks a subset of LH cells. The study is interesting either way, but answers to this question could be important in guiding future studies.

Here are some specific comments:

1. Given that 66% of Penk+ cells GABAergic and 85% of Penk+ cell are glutamatergic. Are there many LH cells co-express Vgat and Vglut2? Please clarify.
2. Figure 2 shows the extent of overlap between Penk, Hcrt and Lepr. The results suggest partial overlap between those genes, but it is not clear why these genes are chosen and whether the overlap should be considered high or low. It will be helpful to quantify the percentage of Penk, Hcrt and Lepr cells in the total population to understand whether the overlap is higher or lower than the chance level.
3. Are LH Penk cells preferentially activated by predator odor? Do the cells also respond to a neutral odor or attractive odor? Furthermore, do the LH Penk- cells respond to predator odor or HFD consumption? Does the PSS +HFD induced c-Fos overlap preferentially with Penk? These questions are related to the reviewer's general comment regarding how special are LH Penk cells in mediating this behavior in comparison to other LH cells. c-Fos data should be presented at the minimum.
4. Why PSS and LH Gq increases consumption of HFD but not NC? This result suggests that LH does not simply drive feeding. Please discuss the implication of the results regarding the function of LH Penk cells in controlling food intake.
5. It is interesting that the authors used Gq for activation but Kir2.1 for inactivation. Is there any specific reason to choose Kir2.1 over Gi?
6. For Figure 5c, please quantify the % of GR+ and Penk+ cells in the LH to better address whether the extent of overlap is higher or lower than chance level.
7. LH GR antagonist does not specifically target LH Penk cells, making it difficult to assess the contribution of GR blocking in LH Penk cells. If the authors want to conclude that Cort specifically target GR expressed in LH Penk cells to modulate the behavior, please knockout/down GR specifically in LH Penk cells.

8. Please discuss the potential cellular and synaptic mechanisms that may cause increased responses of LH Penk cells to HFD after Cort application.
9. For Figure 4e and f, please use 2-way ANOVA for analysis.
10. For Figure 4b, please include a saline control group.

Reviewer #2 (Remarks to the Author):

In this study, You and colleagues employed interdisciplinary techniques to investigate the role of lateral hypothalamus proenkephalin (LH^{Penk}) neurons in driving threat-induced overeating associated with negative emotions. Using the predator-scent stimulus (PSS) paradigm, they observed a delayed behavioral response characterized by increased high-fat diet (HFD) consumption and negative emotional valence. Using genetically modified mice and c-Fos staining, the authors found that LH^{penk} neurons receive input from the paraventricular nucleus (part of the HPA) and are likely involved in emotional/behavioral processing and feeding behavior. In vivo mini scope via gradient index (GRIN) lens was used to record neuronal calcium activities, revealing that LH^{Penk} neurons are activated during HFD consumption after PSS stimulus. Chemogenetic activation of LH^{Penk} neurons using hM3Dq enhanced HFD consumption but not normal chow (NC), while both chemogenetic and optogenetic activation induced aversion. Chronic silencing of LH^{Penk} neurons using Cre-dependent expression of Kir2.1 mitigated the emotional overconsumption associated with a negative emotional state observed in PSS mice. Interestingly, while serum CORT levels were elevated 30 minutes after PSS exposure, they returned to baseline after 24 hours. RNAscope analysis showed that LH^{Penk} neurons highly express glucocorticoid receptors (GR), and CORT injection enhanced HFD consumption after 24 hours. Blocking GR signaling pharmacologically reversed PSS-induced maladaptive changes. This study provides valuable insights into the central mechanism underlying threat-induced overeating associated with negative emotions. The study design is clear, and the results are novel and compelling. The figures are beautifully presented. To further improve the paper, the authors may consider addressing the following comments.

1. The description of the methods, especially on food intake, should have more details. Are all animals singly housed? Single housing causes stress in animals. How long were these animals singly housed before performing PSS? It is unclear if both NC and HFD were both presented at the same time, and then food intake experiments were done, e.g. Fig. 1 f&g. Fig. 4 b&c. If indeed both NC and HFD were presented, it is important to show if NC only food intake will be increased after PSS.
2. How long will the PSS-induced overconsumption of HFD last? Only one day or longer?
3. The authors performed in vivo miniscope imaging of neuronal activities in this study. What are the responses of these LH^{Penk} neuronal responses during PSS?

4. The author claimed that the LH^{Penk} neurons receive input from the PVH glutamatergic input, and the involvement of LH^{Penk} in overfeeding after PSS is downstream of PVH (Page 18). Do PVH-origin synaptic inputs required for the activation of these LH^{Penk} neurons during HFD consumption?

5. The authors tried to clarify the neurochemistry of the LH^{Penk} neurons using FISH. The data presented in Figure 2 indicates that LH^{Penk} neurons are heterogeneous. Notably, while the majority of the MCH and hypocretin neurons do not express Penk, the majority of glutamatergic (84.79%) and GABAergic (66.51%) neurons express Penk. While this is clear, it is difficult to know whether high-fat diet (HFD) consumption after PSS is primarily driven by the release of glutamate/GABA from the LH or Penk peptides. Moreover, it would be informative to show what percentage of Penk neurons are glutamatergic and what percentage of Penk neurons are GABAergic (i.e. by quantifying Vgat+/Penk+ and Vglut2/Penk+) within the hypothalamic region.

6. Chemogenetic activation of LH^{Penk} neurons acutely enhances HFD consumption. It would be interesting to explore whether optogenetic activation of these neurons also induces HFD consumption (e.g. Supplemental Figure 5). Additionally, it is important to consider the differences between the acute modulation of LH^{Penk} neurons inducing HFD consumption and the 24-hour delay required for CORT (stress) to have a similar effect. Does the activation of LH^{Penk} neurons drive overconsumption of NC?

7. Kir2.1-mediated inhibition of LH^{Penk} neurons alleviated emotional overconsumption associated with negative emotional states in PSS mice. It would be interesting to investigate whether chronic manipulation of these neurons (in this case, inhibition of LHA^{Penk} neurons) impacts the animals' body weight or metabolism.

Reviewer #3 (Remarks to the Author):

In this paper they identified a population of proenkephalin-expressing LH neurons that increase engagement during high-fat diet eating 24 hours after a stress response-inducing exposure to predator odor and showed that chemogenetic activation of LHPenk neurons is sufficient to drive HFD overconsumption in an unstressed animal while also promoting a negative valence state as evidenced by formation of a conditioned place aversion. These findings suggest that stress-induced modulation of LHPenk activity via a GR-dependent mechanism induces negative affect to promote overconsumption of highly palatable food. Overall, the paper makes a compelling case for the GR-dependent mechanisms, but the lack of consideration for a role of Penk failed to directly link the genetic ID of the cells with anything meaningful, and simply treating the Enk as a marker is

insufficient to support publication at this time. More evidence for specific role of the Enk would substantially improve the manuscript.

Major Considerations:

1. The role of enkephalin as a marker for this LH neuron population remains unclear. There is nothing which suggest that all LH neurons are not involved in the behavior; they show that the Penk LH population has little overlap with other prominently studied LH populations (Pmch, Hcrt, Lepr), but there aren't any control experiments showing that these other populations don't have the same increase in cFos during HFD after stress/activity during HFD first bout, etc. I think the easiest way to start getting at this would be in Figure 6j-k, what proportion of cFos+ neurons are also GFP-? We only get a quantification of Penk cells that are cFos+, it would be more convincing to show that non-Penk neurons in the LH are not activated by this condition. Otherwise, it feels like we are arbitrarily winnowing in on a 'subpopulation' without evidence that any of these outcomes are unique. Since there is also no data showing a functional role for enkephalin in this context (see point 2), I think there needs to be more evidence for why we think Penk is the marker here.

2. Any evidence of a functional role for LH enkephalin in stress-related binge-eating should be provided in at least one or two ways (to be rigorous). Since Penk is used as the marker identifying this population, this paper would be substantially more impactful and stronger if they can demonstrate that enkephalin is involved in these behaviors. If it is not, following experimental manipulations, then the authors will need to demonstrate why this particular population is special as compared to other LH populations (see above pt 1).

Some experimental ideas which would facilitate the above:

a. Conditional knockout of Penk in the LH, this gives you multiple options for evaluating the functional role of enkephalin in this context. Test if PSS still produces higher HFD consumption and CPA. Test if PSS still increases cFos expression in the hypothalamus. Test if LHPenk calcium activity is still elevated during the first HFD eating bout after stress. These will start answering if enkephalin has a role in the process, and if so at what point in the process. Is enkephalin involved in the sensitization of these neurons to stress during the actual stress exposure? Is it released during the consumption period post stress? Is it relevant in HFD and NC consumption or only HFD? Feasibility of CRISPR guide to induce Cre-dependent Penk cKO demonstrated in Castro Nature 2021.

b. Use the already obtained input/output tracing data to identify the pre/ post-synaptic opioid receptor population that these LH neurons would release enkephalin onto. This would open up many doors to test a functional enkephalin role (e.g. local infusion of MOPR agonist/antagonist in sufficiency and necessity of driving the PSS valence and consumption responses, selective knockout of receptor in the downstream population, record calcium activity of the downstream MOPR population during HFD eating post-stress to see if there are changes in activity corresponding with the increases in LHPenk activity during the first feeding bout). I think this would also be the strongest narrative-wise, to be able to isolate endogenous opioid function and to also map a more complete circuit driving the behavior.

c. The simplest starting experiment: does treatment with an opioid receptor antagonist either before stress or before post-stress feeding prevent any of the phenotypes? This could at least be a starting point for identifying any opioid function in stress binge eating before spending too much time with the more careful experiments.

3. The mifepristone microinfusion experiment in Figure 6 is somewhat unsatisfying in the sense that the antagonist will block all GRs in the LH and not just LHPenk GRs. How large is the GR+ Penk-population (this should be quantifiable from the dataset in 5c)? If small, then this is maybe less of a concern. If there are a significant number of non-Penk LHGR neurons however, then I think a follow-up experiment completing a conditional knockout of GR on LHPenk neurons is warranted.

4. In Figure 3, what is the explanation for using a novel object as the control for HFD instead of the NC food, which is used throughout the rest of the paper?

5. The effect in Figure 5j of a CORT-induced increase in LHPenk activity during 1st bout HFD feeding appears to be predominantly driven by 3 cells – it is unclear if this is spread across all 4 animals that were recorded or if these cells are from one animal. I think it would be more convincing to add a few more animals to these groups.

6. I'm unsure about using conditioned place aversion as a metric for persistent 'negative emotional valence.' When thinking about this in a real-world context, a sustained negative emotional state following a stressful event does not require re-exposure to the context where the stressful event occurred. To me the CPA is indicative of learning and memory as the mice formed an association between the context and the stimulus, as opposed to the authors interpretation of CPA as a retention of negative emotions. Would those negative emotions exist if they weren't exposed to the stressful context again? The authors might consider other models of negative affect to bolster their results and conclusions.

7. What was the justification for using Kir2.1 as a chronic silencing method in Figures 4h-j? It would seem more compelling to use a more temporally-precise silencing tool (e.g., halorhodopsin) to identify the relevant times where LHPenk activity is necessary to induce HFD overconsumption.

8. There needs to be a greater distinction between what the authors refer to as "negative emotion" and fear memory. How can they be sure that their results are due to LHPenk involvement in negative emotions and not some other aspect of aversion? This is a related point to the one above about negative affect/valence.

9. What is the point of the tracing/circuit data? This data feels out of place in the paper and largely ignored throughout the rest of the experiments. The results from the in situ hybridization experiments are also largely ignored in the remainder of the text. How these results help the authors

gain insight into the role of LHPenk neurons remains unclear. Rearranging the order in which the data is presented might help address these issues.

10. Does the GR antagonist block the effects of corticosterone on LHPenk activity following PSS and during GFD consumption? The only experiment done using a combination of the GR antagonist and corticosterone focused on their effects on consumption, not neuronal activity.

Minor Considerations:

-The introduction is missing a discussion of any previous investigations of LHPenk existence/function, and only talks about enkephalin's role in other brain regions. What evidence is there that Penk expresses in the LH, what is known already about these neurons?

-The LH is a neuroanatomical hub responsible for regulating diverse primitive behavioral states, such as feeding, as well as approach and avoidance behaviors." This claim on page 8 is missing several citations. Primarily Nieh et al. Neuron, 2016, and Marino et al., Proc Natl Acad Sci USA, 2020, which show LH circuits involved in food approach, negative affect, approach/avoidance behavior.

-Switch the color scheme on the pie charts in Figure 2 so that the +/+ group is always in color, and the +/- group is always in grayscale. It's confusing to swap what corresponds with colored vs grayscale between 2h and 2j, and I think the default when glancing at a figure is to assume the group in color is the positive condition (in this case, both markers co-localize) and the grayscale condition is the 'null' (Penk does not co-localize with the other marker). I think that will highlight the idea that the Penk neurons are mainly a standalone population, regardless I think it should be consistent throughout this figure and not switch once you get to Vgat/Vglut2.

-“Consumption of palatable foods can activate neural activity of the hypothalamus” (Page 7). Is this claim meant in a no-stress context? If so, the results in Extended Data Figure 1 do not agree with this statement, where HFD did not increase neural activity in LH, PVN, DMH, VMH, or ARC relative to NC diet in the unstressed mice.

-For the anterograde and retrograde tracing experiments in Extended Data Figure 2, can the authors show the negative data that the DIO viral injections were contained specifically to the LH? The data in Figure 2a suggest that there are other Penk-expressing hypothalamic subregions, so the tracing could potentially pick up inputs/outputs not specific to the LH if the viral expression was not restricted.

-Can the author's show individual data for Figures 4e-g? The medians appear identical for the hM3D group pre/post-test.

-In Figures 5f-g, using L and H to denote the low and high doses of mifepristone is overly confusing, especially since this notation isn't used in the main results text. Just say 1 and 10 ng on the figure.

-Does mifepristone treatment prior to PSS have an effect on HFD consumption immediately following PSS? Since PSS only had an effect on delayed HFD and not immediate HFD (Figure 1f), it would be

more convincing that GR is the responsible mechanism for this effect if the authors can show that mifepristone also does not impact HFD if assessed immediately after PSS.

-In Extended Data Figures 6m-n, the authors should add a CORT NC control group to show that CORT doesn't just generally increase LHPenk activity, and that the effect of the CORT HFD group is actually specific to the HFD.

-Page 10, Line 1 : Replace "This led us..." with "These results led us..."

-Figure 4i+j , 6c+d : Symbols depicting the data points are empty instead of filled in like on all other figures. Choose between filled in or empty for all graphs.

-Extended Data Fig. 5g : Squares are empty instead of filled in compared to 5h depicting the same data.

Reviewer #1 (Remarks to the Author):

In this study, You et al. demonstrated LH Penk cells as a critical population for controlling stress-induced overeating. They showed that predator cues induced acute anxiety, conditioned avoidance and overeating of palatable food 24 hours later. The authors then demonstrated LH Penk cells can bi-directionally modulate the HFD overeating and conditioned avoidance. In vivo calcium recording further demonstrated an increased activity of LH Penk cells to HFD after predator odor exposure. The authors went one step further to demonstrate that Cort is an important mediator for the predator odor induced HFD overeating. Blocking GR in LH can block Cort or predator odor induced HFD overeating. The study is well performed and novel. However, the reviewer was left uncertain whether LH Penk cells are indeed unique in mediating the predator odor-induced overeating or Penk simply marks a subset of LH cells. The study is interesting either way, but answers to this question could be important in guiding future studies.

We thank the Reviewer for their supportive comments about our manuscript. We also appreciate the Reviewer for the time and effort they allocated to providing a thorough review aimed at improving the paper.

Here are some specific comments:

1. Given that 66% of Penk+ cells GABAergic and 85% of Penk+ cell are glutamatergic. Are there many LH cells co-express Vgat and Vglut2? Please clarify.

We have performed fluorescent *in situ hybridization* (FISH) experiments and investigated VGAT and vGlut2 expression in the LH. We found that VGAT- and vGlut2-rich subregions are somewhat segregated or intermingled within the whole LH, indicating 50.78%, 47.02% of VGAT- and vGlut2-positive LH neurons, respectively, but only a small subset (2.20%) co-expressed VGAT and vGlut2 (**please see figures below**; Scale bars, 250 μ m). In our original manuscript, we quantified Penk-positive fractions in each VGAT- and vGlut2-rich LH subregions separately and found that Penk-expressing neurons are enriched in both populations.

To fully clarify the Reviewer's point, we performed additional FISH experiments with the new combination of probes including Penk (C1), VGAT (C2) and vGlut2 (C3), altogether. We then quantified what % of Penk neurons are glutamatergic or GABAergic *within the whole LH* including the VGAT- and vGlut2-rich LH subregions. We found that about 52.65%, 42.42% of Penk+ cells co-expressed vGlut2 and VGAT, respectively, and 4.92% of them expressed on both VGAT- and vGlut2-expressing subpopulation (**Fig. 2i, j and Extended Data Fig. 2d in the revised manuscript**).

We presented the updated data in **Fig. 2i, j and Extended Data Fig. 2d of the revised manuscript**. We hope that we have now clarified the Reviewer's point.

2. Figure 2 shows the extent of overlap between Penk, Hcrt and Lepr. The results suggest partial overlap between those genes, but it is not clear why these genes are chosen and whether the overlap should be considered high or low. It will be helpful to quantify the percentage of Penk, Hcrt and Lepr cells in the total population to understand whether the overlap is higher or lower than the chance level.

We have chosen Hcrt and Lepr because they could have somewhat similar functional relevance to the Penk in controlling behavioral stress responses and palatable food consumption. Specifically, Hcrt-expressing LH neurons are known to be

sufficient to activate the hypothalamic–pituitary–adrenal (HPA) axis and release corticosterone (CORT), whereas a leptin system is known to antagonize the Hcrt-mediated stress responses and attenuate the HPA-axis activation¹. Furthermore, Lepr-expressing LH neurons are involved in the central regulation of palatable food overconsumption, especially under intermittent-access feeding paradigms². These functions of Hcrt and Lepr in the LH could be related to the role of LH^{Penk} neurons in promoting the consumption of palatable foods primarily through CORT-mediated behavioral stress responses. Therefore, we wanted to examine whether the LH^{Penk} neurons represent a discrete population compared with Hcrt- or Lepr-expressing LH neurons.

As requested, we also have performed additional FISH experiments and quantified the % of Penk, Hcrt and Lepr cells in the *total LH population*. Applying mixtures of three target probes, including Penk (C1), Hcrt (C2) and Lepr (C3), we found that the majority of LH^{Penk} neurons (76.44%) are stand-alone, but lower proportions of LH^{Penk} neurons also express Hcrt (6.12%) and Lepr (17.07%) (**Extended Data Fig. 2a-c in the revised manuscript**), consistent with our previous data in **Fig 2e-h in the revised manuscript**.

We have added the related descriptions in the Results section (**page 8**) and new data in **Extended Data Fig. 2a-c in the revised manuscript**.

3. Are LH Penk cells preferentially activated by predator odor? Do the cells also respond to a neutral odor or attractive odor? Furthermore, do the LH Penk- cells respond to predator odor or HFD consumption? Does the PSS +HFD induced c-Fos overlap preferentially with Penk? These questions are related to the reviewer's general comment regarding how special are LH Penk cells in mediating this behavior in comparison to other LH cells. c-Fos data should be presented at the minimum.

This point is well taken. In our original manuscript, we showed that LH^{Penk} neurons showed an increase in the number of Ca²⁺ transients when the mice were exposed to PSS, which returned to baseline after the PSS removal (**Extended Data Fig. 4a-k in the revised manuscript**). However, in the presence of an empty container with the neutral odor only, the LH^{Penk} neurons exhibited no substantial changes in activity (**Extended Data Fig. 4l-n in the revised manuscript**), suggesting that LH^{Penk} neuronal activity preferentially encodes the PSS-driven negative emotional state.

We also agree that knowing whether the *c-fos* expression induced by PSS+HFD condition preferentially overlaps with LH^{Penk} neurons would allow us to make more conclusive statements. We therefore asked whether the HFD-induced expression of *c-fos* observed in PSS mice is mainly occurring in the LH^{Penk} neurons compared to Lepr-expressing LH (LH^{Lepr}) neuronal population representing one of the non-Penk LH subsets (**Fig. 2g, h, and Extended Data Fig. 2a-c in the revised manuscript**). To do this, we injected AAV expressing a Cre-dependent eGFP (AAV-DIO-eGFP) into the LH of Penk-Cre or Lepr-Cre mice to label the Penk- and Lepr-positive LH population, respectively. We then counted LH cells showing both *c-fos* immunoreactivity and eGFP fluorescence. After the PSS-induced HFD overconsumption, 29.91% of Penk-positive LH cells contained detectable *c-fos*+ nuclei; To a lesser extent, only 10.72% of the Lepr-positive LH cells showed *c-fos* expression under the same condition of PSS+HFD (**Fig. 5f-h in the revised manuscript**). This supports our hypothesis that the LH^{Penk} neurons constitute a relevant population preferentially activated by PSS-induced HFD overconsumption.

In addition, to substantiate our ideas that Penk in the LH is *functionally unique in mediating the PSS-induced overeating* (instead of being a simple marker for subsets of LH cells), we downregulated Penk signaling in the LH circuit and examined whether LH endogenous enkephalin is necessary for PSS-induced maladaptive changes at the neuronal and behavioral levels.

First, we injected an adeno-associated virus (AAV) carrying a Cre-dependent short hairpin RNA (shRNA) against Penk (AAV-DIO-EmGFP-Penk shRNA) into the LH of Penk-Cre mice and confirmed a robust reduction in Penk mRNA expression without accompanying changes in Hcrt, Pmch, and Lepr (**Fig. 5a, b in the revised manuscript**). In response to HFD exposure at 24 h post-PSS exposure, mice expressing Penk shRNA in the LH consumed much lesser amounts of HFD, compared to PSS mice expressing control fluorescence (AAV-DIO-EmGFP) (**Fig. 5c, d in the revised manuscript**). Notably, under the PSS+HFD condition, the shRNA-mediated knockdown of Penk in the LH reduced overall *c-fos* induction, particularly in the of LH^{Penk} neuronal population (**Fig. 5f-h in the revised manuscript**). These data suggest that the

disruption of endogenous enkephalin in the LH is critical for inhibiting various features we observed in PSS mice, such as HFD overconsumption and sensitized responsiveness of LH^{Penk} neurons upon HFD exposure.

Second, to further confirm our shRNA data, we also applied a CRISPR/SaCas9 viral-based system. AAV carrying a sgRNA targeted to the mouse Penk locus along with *Staphylococcus aureus* Cas9 (SaCas9; AAV-CMV-FLEX-SaCas9-U6-sgPenk³; Addgene #159916) was bilaterally injected into the LH of Penk-Cre mice. We confirmed that CRISPR infection reduced Penk expression by 70% and behaviorally suppressed PSS-induced HFD overconsumption, whereas control PSS mice with intact Penk production showed increased HFD intake after PSS exposure (**Fig. 5i-k in the revised manuscript**). Furthermore, CRISPR-mediated deletion of Penk has a critical role in protection against PSS-induced negative emotional valence in the CPA experiment (**Fig. 5m, n in the revised manuscript**), demonstrating that enkephalin peptides in the LH are necessary for mediating PSS-driven negative emotional state associated with HFD overconsumption. Taken together, these additional results further support that the Penk-expressing LH cells play unique roles in eliciting maladaptive behavioral changes in the context of PSS.

Third, we went one step further to investigate the receptor target of endogenous enkephalin in the LH. Because Penk is cleaved into potent endogenous peptide agonists acting on μ -opioid peptide receptors (MORs) that are known to modulate binge eating behaviors⁴, we hypothesized that activation of MOR signaling in the LH^{Penk} neuronal efferent projection is a critical component that relays information from LH^{Penk} neurons, leading to HFD overconsumption. The lateral periaqueductal gray (LPAG), one of the downstream structures of LH^{Penk} neurons, highly expresses MORs⁵, we thus tested whether HFD overconsumption induced by artificial activation of LH^{Penk} neurons is affected by inhibition of MOR activity in the LPAG. Indeed, while the mice expressing hM3D in LH^{Penk} neurons showed significantly enhanced HFD consumption in the presence of CNO, intracerebral microinjections of the MOR-selective antagonist CTAP (0.1 or 1 μ g/side) into the LPAG under the same condition reduced HFD intake levels in a dose-dependent manner (**Fig. 6d-f in the revised manuscript**). Likewise, we found that a local infusion of the CTAP (1 μ g/side) into the LPAG normalized HFD consumption at 24 h post-PSS exposure (**Fig. 6i, j in the revised manuscript**), suggesting that MOR signaling in the LH^{Penk}-LPAG circuit primarily governs HFD overconsumption following PSS exposure.

Overall, these data indicate the functional relevance of the endogenous enkephalinergic systems in the LH circuitry in modulating post-threat effects on emotional overeating, supporting the ideas that LH^{Penk} neurons are indeed unique in mediating PSS-induced detrimental effects.

All of these new data are now presented in **Fig. 5, 6 in the revised manuscript**. We hope that we have now addressed the Reviewer's concern.

4. Why PSS and LH Gq increases consumption of HFD but not NC? This result suggests that LH does not simply drive feeding. Please discuss the implication of the results regarding the function of LH Penk cells in controlling food intake.

We agree that this is an important point to discuss further. Although decades of research suggest that modulation of LH neurons leads to up- or down-regulation of feeding behaviors, very little is known about the cellular/circuit specificity of the LH and its functional roles in emotional eating behavior, especially *tendency to eat palatable foods*. Particularly, coping with stressful situations can boost *hedonic hunger* triggered by psychological needs (instead of actual needs from physical hunger), resulting in increased motivation to consume foods, rich in sugar and fat (so called 'comfort eating').

Previous studies suggested that stress changes the activity of the HPA axis, which influences hormonal/neuropeptidergic systems in the hypothalamus that regulates appetite and *hedonic aspects of eating*, leading to cravings for palatable food^{6,7}. In this manuscript, we identified LH^{Penk} neurons as a critical neural component that specifically promotes palatable food consumption, as demonstrated by the fact that activation of LH^{Penk} cells increases HFD consumption without accompanying changes in general NC intake (**Fig. 4a-c in the revised manuscript**). In the PSS context, we speculate that LH^{Penk} cells are likely to be a primary target that is permissive for the induction of hedonic feeding after interacting with appetite-stimulating stress hormone, corticosterone (CORT) (**Fig. 7 in the revised manuscript**). We believe our discovery of LH^{Penk} neuron-mediated overeating targeted selectively to palatable foods presents a previously unknown mechanism by which the threat-induced behavioral stress responses influence the hypothalamic neuropeptidergic systems that control emotionally triggered eating, distinct from homeostatic eating.

We have added the related description in the **Discussion section (p.21) of the revised manuscript**.

5. It is interesting that the authors used Gq for activation but Kir2.1 for inactivation. Is there any specific reason to choose Kir2.1 over Gi?

PSS-induced HFD overconsumption requires time to take hold as evidenced in our data showing that mice exhibited HFD overconsumption 24 h after, but not immediately after the PSS exposure (**Fig. 1f, g in the revised manuscript**). Moreover, CORT-driven HFD overconsumption is clearer a day after, but not immediately after the CORT administration (**Extended Data Fig. 7b in the revised manuscript**), suggesting that time delay between the PSS-induced CORT elevation and subsequent HFD exposure would allow for mice to develop and display their heightened craving for HFD eating. We speculated that LH^{Penk} neurons may be the key component that has undergone CORT-induced neuronal adaptation during the delay between the PSS exposure and the measurement of HFD intake. It is also likely that any active adaptive processes activating LH^{Penk} neurons occur during this interval.

We performed chronic silencing experiments with Kir2.1 instead of Gi, in order to block the potential LH^{Penk} neuronal activity changes during the interval (24 h) without the information on *when* the maladaptive responses of PSS-primed LH^{Penk} neurons will be expressed. Afterwards, through real-time monitoring of *in vivo* LH^{Penk} neuronal activity, we observed that the increased activity of PSS-primed LH^{Penk} neuronal activity is prominent upon HFD exposure.

6. For Figure 5c, please quantify the % of GR+ and Penk+ cells in the LH to better address whether the extent of overlap is higher or lower than chance level.

As requested, we revisited FISH data for GR+ and Penk+ cells and extended the quantification area into total LH population, instead of focusing on Penk-rich LH subregions. Within the whole LH - defined as lateral and up to 0.1 mm medial to the fornix or up to 0.2 mm above or below the fornix - we found that the majority of Penk+ neurons (79.48%) are GR+, as evidenced by co-expression of Penk and GR, while 19.36% of total GR+ neurons expressed in the Penk+ neurons. We have added the updated data in **Fig. 7c, d in the revised manuscript**.

7. LH GR antagonist does not specifically target LH Penk cells, making it difficult to assess the contribution of GR blocking in LH Penk cells. If the authors want to conclude that Cort specifically target GR expressed in LH Penk cells to modulate the behavior, please knockout/down GR specifically in LH Penk cells.

We agree on this point. We have designed and tested several AAV constructs expressing shRNA against GR, but unfortunately, none of those constructs showed relevant knockdown potency. Instead of applying knockout/down GR, we thus planned to *overexpress* GR *specifically* in LH^{Penk} neurons and tested whether the overexpression of GR in LH^{Penk} cells can abolish GR antagonist (mifepristone)-induced behavioral normalization. To test this, a mouse cDNA for GR (Nr3c1; NM 008173.4) was amplified by PCR and inserted into vector to produce AAV-hsyn-DIO-GR. We stereotaxically injected the AAV-hsyn-DIO-GR into LH of Penk-Cre mice and confirmed that GR mRNA levels were robustly increased using both FISH experiments and quantitative real-time PCR (qRT-PCR) assay (**Fig. 7j and Extended Data Fig. 7f in the revised manuscript**). We indeed found that LH^{Penk} neuron-specific overexpression of GR prevented microinfusion of mifepristone from inhibiting the CORT-induced HFD overconsumption (**Fig. 7k in the revised manuscript**). This confirms that the effects of mifepristone on normalizing HFD intake are primarily mediated by inhibiting GR in the LH^{Penk} cells.

We presented the new data in **Fig. 7i-k and Extended Data Fig. 7f of the revised manuscript**. We hope that we have now addressed the Reviewer's concern.

8. Please discuss the potential cellular and synaptic mechanisms that may cause increased responses of LH Penk cells to HFD after Cort application.

Corticosterone (CORT) is known to have an ability to change synaptic plasticity in several brain regions, leading to profound effects on behavioral stress responses. Previous studies identified α -amino-3-hydroxy-5-methyl-4-

isoxazolepropionic acid receptor (AMPA) as a major target of this process, because CORT can increase AMPAR-mediated synaptic transmissions via rapid/non-genomic and slow/genomic mechanisms^{8,9}. Based on our data showing that CORT pretreatment increased the responsiveness of the LH^{Penk} neurons to HFD consumption (**Fig. 7I-n and Extended Data Fig. 8 in the revised manuscript**), we speculate that CORT may increase the surface expression and synaptic localization of AMPAR subunits in the LH^{Penk} neurons, potentially leading to enhanced glutamatergic synaptic transmission. It is also plausible that those processes are mediated by slow/genomic mechanisms involving transcriptional regulation of GR-affected genes, because CORT administration leads to delayed onset of emotional eating (**Extended Data Fig. 7b in the revised manuscript**). Particularly, given that GR activation can alter genes functionally involved in regulating GluR2/3 delivery and/or membrane anchoring, future studies into the CORT-mediated genomic processes in the LH and multiple candidate genes that could directly influence AMPA receptor trafficking in LH^{Penk} neurons will further our understanding of cellular/synaptic mechanisms underlying delayed-onset emotional overeating after PSS exposure.

We have added the related description in the **Discussion section (p.24) in the revised manuscript**.

9. For Figure 4e and f, please use 2-way ANOVA for analysis.

As requested, we reanalyzed the data in **Fig. 4e, f** with Two-way RM ANOVA ($F_{(1,16)} = 5.011$, $p = 0.040$) followed by Bonferroni post hoc test to compare the results in pre vs. post x AAV-DIO-mCherry vs. AAV-DIO-hM3D. We also performed additional experiments with three AAV-DIO-mCherry mice and two AAV-DIO-hM3D mice, thereby increasing the total n number up to 9 mice per group. We confirmed that CNO-mediated LH^{Penk} neuronal activation induced strong conditioned place aversion, strengthening our previous conclusion [$*p = 0.016$ for AAV-DIO-hM3D: pre- vs. post-test; $\dagger p = 0.015$ for post-test: AAV-DIO-mCherry vs. AAV-DIO-hM3D].

We presented the updated data in **Fig. 4e of the revised manuscript**.

10. For Figure 4b, please include a saline control group.

We have included saline control data in **Extended Data Fig. 6a, b in the revised manuscript**.

Reviewer #2 (Remarks to the Author):

In this study, You and colleagues employed interdisciplinary techniques to investigate the role of lateral hypothalamus proenkephalin (LH^{Penk}) neurons in driving threat-induced overeating associated with negative emotions. Using the predator-scent stimulus (PSS) paradigm, they observed a delayed behavioral response characterized by increased high-fat diet (HFD) consumption and negative emotional valence. Using genetically modified mice and c-Fos staining, the authors found that LH^{Penk} neurons receive input from the paraventricular nucleus (part of the HPA) and are likely involved in emotional/behavioral processing and feeding behavior. In vivo mini scope via gradient index (GRIN) lens was used to record neuronal calcium activities, revealing that LH^{Penk} neurons are activated during HFD consumption after PSS stimulus. Chemogenetic activation of LH^{Penk} neurons using hM3Dq enhanced HFD consumption but not normal chow (NC), while both chemogenetic and optogenetic activation induced aversion. Chronic silencing of LH^{Penk} neurons using Cre-dependent expression of Kir2.1 mitigated the emotional overconsumption associated with a negative emotional state observed in PSS mice. Interestingly, while serum CORT levels were elevated 30 minutes after PSS exposure, they returned to baseline after 24 hours. RNAscope analysis showed that LH^{Penk} neurons highly express glucocorticoid receptors (GR), and CORT injection enhanced HFD consumption after 24 hours. Blocking GR signaling pharmacologically reversed PSS-induced maladaptive changes. This study provides valuable insights into the central mechanism underlying threat-induced overeating associated with negative emotions. The study design is clear, and the results are novel and compelling. The figures are beautifully presented. To further improve the paper, the authors may consider addressing the following comments.

We thank the Reviewer for the time spent on understanding our manuscript and the enthusiastic assessment of our data. We address each suggestion for additional improvements below.

1. The description of the methods, especially on food intake, should have more details. Are all animals singly housed? Single housing causes stress in animals. How long were these animals singly housed before performing PSS? It is unclear if both NC and HFD were both presented at the same time, and then food intake experiments were done, e.g. Fig. 1 f&g. Fig. 4 b&c. If indeed both NC and HFD were presented, it is important to show if NC only food intake will be increased after PSS.

There was *no* single-housing period *before* performing PSS. It only occurs either immediately or 24 h *after* PSS exposure in order to assess food intake levels. Under this single housing condition, mice had *ad libitum* access to the NC and HFD *at the same time* thus receiving 2.5 h free-choice access to *both* foods (**Fig. 1f, g in the revised manuscript**). Likewise, for chemogenetic activation experiments (**Fig. 4b, c in the revised manuscript**), mice were individually housed after intraperitoneal injections of CNO (2 mg/kg) and then given free-choice access to the NC and HFD. The food intake measurements were taken 2.5 h after the drug administration. We have added more details in **the Methods under 'Food intake measurements' section of the revised manuscript**.

As requested, we have measured food intake levels under *NC only* condition. Given that NC was always available *ad libitum*, mice were less likely to initiate active eating bouts for the NC during the general time window (2.5 h). We therefore extended food monitoring time up to 24 h after PSS exposure and examine whether PSS affects food intake level under the NC only condition, with no food-choice access. We found that PSS mice showed no substantial difference in NC intake levels (**please see a figure below**), confirming that PSS selectively triggered emotional overeating toward HFD without accompanying changes in general NC intake [Two-tailed unpaired *t*-test; $t_{11} = -1.126$, $p = 0.284$]. As we responded to the Reviewer 1's comment # 4, we speculate that PSS exposure preferentially elevated the susceptibility to 'palatable food' overconsumption via activating LH^{Penk} neurons which may play an important role in increasing hedonic eating, leading to craving for foods, rich in sugar and fat.

We have added the related description in the **Discussion section (p.21) of the revised manuscript**.

2. How long will the PSS-induced overconsumption of HFD last? Only one day or longer?

In our original pilot study, we performed HFD intake measurements with three different time points: immediately, 24 h, and 72 h after PSS exposure. We found the highest HFD consumption at 24 h post-PSS exposure but a moderate level of consumption at 72 h time mark (**please see data below, left**) [One-way ANOVA was followed by Fisher LSD post hoc test for multiple comparisons; *** $p < 0.001$, † $p = 0.023$ compared with 24 h Post-PSS mice]. Based on this result, we believe that the effects of PSS on HFD overconsumption are most impactful 24 h after the PSS exposure and then weaken over time. In addition, we were not able to observe significant changes in body weight gain of PSS mice under the chronic HFD exposure for four days (**please see data below, middle and right**), suggesting that a single PSS exposure is likely to be most effective to increase HFD intake within a certain time window, which is 24 h.

3. The authors performed in vivo miniscope imaging of neuronal activities in this study. What are the responses of these LH^{Penk} neuronal responses during PSS?

In our original manuscript, we claimed that LH^{Penk} neurons showed an increase in the number of Ca²⁺ transients *during* PSS, which returned to baseline after PSS removal (**Extended Data Fig. 4a-c in the revised manuscript**). However, in the presence of an empty container with the neutral odor only, the LH^{Penk} neurons exhibit no substantial changes in activity (**Extended Data Fig. 4l-n in the revised manuscript**), suggesting that LH^{Penk} neuronal activity preferentially encodes the PSS-driven negative emotional state.

4. The author claimed that the LH^{Penk} neurons receive input from the PVH glutamatergic input, and the involvement of LH^{Penk} in overfeeding after PSS is downstream of PVH (Page 18). Do PVH-origin synaptic inputs required for the activation of these LH^{Penk} neurons during HFD consumption?

In the Discussion section of our original manuscript, we raised the possibility that PSS-primed PVN can be a critical upstream structure that sensitizes LH^{Penk} neuronal reactivity to HFD, based on our input tracing data (**Extended Data Fig. 3 in the revised manuscript**) as well as *c-fos* data (**Extended Data Fig. 1a-d in the revised manuscript**). Since the PVN to LH^{Penk} neuronal connection has been poorly defined, this will be interesting topics and important in guiding future studies. Applying a simultaneous optogenetics and *in vivo* Ca²⁺ imaging platform (nVoke, Inscopix, Palo Alto, California), we also performed pilot studies and produced preliminary data indicating the functional connection of PVN- LH^{Penk} neurons. In this revised manuscript, however, we believe that this topic is beyond the scope, because we have been focusing on the role of LH^{Penk} neurons as well as endogenous enkephalinergic systems in the LH-LPAG circuitry.

5. The authors tried to clarify the neurochemistry of the LH^{Penk} neurons using FISH. The data presented in Figure 2 indicates that LH^{Penk} neurons are heterogeneous. Notably, while the majority of the MCH and hypocretin neurons do not express Penk, the majority of glutamatergic (84.79%) and GABAergic (66.51%) neurons express Penk. While this is clear, it is difficult to know whether high-fat diet (HFD) consumption after PSS is primarily driven by the release of glutamate/GABA from the LH or Penk peptides. Moreover, it would be informative to show what percentage of Penk neurons are glutamatergic and what percentage of Penk neurons are GABAergic (i.e. by quantifying Vgat+/Penk+ and Vglut2/Penk+) within the hypothalamic region.

In our revised manuscript, we updated the Penk/glutamatergic/GABAergic neuronal quantification data, as we responded to the Reviewer 1's comments #1; We performed additional FISH experiments by applying three RNAscope[®] probes altogether (Penk-C1/VGAT-C2/vGlut2-C3) and quantified % of Penk neurons that are glutamatergic or GABAergic within the whole LH, instead of quantifying within the VGAT- or vGlut2-rich LH subregions separately. We found that about 42.42%, 52.65% of Penk LH neurons co-expressed either VGAT or vGlut2, respectively, and 4.92% of them expressed on both VGAT- and vGlut2-expressing subpopulation in the LH. We presented the updated data in **Fig. 2i, j and Extended Data Fig. 2d of the revised manuscript**.

Regarding the question *whether high-fat diet (HFD) consumption after PSS is primarily driven by the release of glutamate/GABA from the LH or Penk peptides*, we have performed new experiments investigating the functional roles of Penk peptides in the LH. *First*, to test the necessity of LH Penk in potentiating HFD consumption after PSS exposure, we applied a CRISPR/SaCas9 viral-based system and disrupted enkephalin in the LH cells (**Fig. 5i, j in the revised manuscript**). At 24 h post-PSS exposure, we observed that CRISPR-mediated deletion of Penk in the LH significantly suppressed PSS-induced HFD overconsumption (**Fig. 5k in the revised manuscript**). Moreover, using shRNA-mediated Penk knockdown, we further confirmed that downregulation of Penk peptide in the LH plays a significant role in preventing HFD overconsumption following PSS exposure (**Fig. 5a-d in the revised manuscript**), consistent with our data using a CRISPR/SaCas9 viral-based system. *Second*, because Penk is cleaved into potent endogenous peptide agonists acting on μ -opioid peptide receptors (MORs) that are known to modulate binge eating behaviors⁴, we hypothesized that activation of MOR signaling in the LH^{Penk} neuronal efferent projection is a critical component that relays information from LH^{Penk} neurons, leading to HFD overconsumption. The lateral periaqueductal gray (LPAG), one of the downstream structures of LH^{Penk} neurons, highly expresses MORs⁵, we thus tested whether HFD overconsumption induced by artificial activation of LH^{Penk} neurons is affected by inhibition of MOR activity in the LPAG. Indeed, while the mice expressing hM3D in LH^{Penk} neurons showed significantly enhanced HFD consumption in the presence of CNO, intracerebral microinjections of the MOR-selective antagonist CTAP (0.1 or 1 μ g/side) into the LPAG reduced the HFD intake levels in a dose-dependent manner (**Fig. 6d-f in the revised manuscript**). Likewise, we found that a local infusion of the CTAP (1 μ g/side) into the LPAG normalized HFD consumption at 24 h post-PSS exposure (**Fig. 6i, j in the revised manuscript**), suggesting that MOR signaling in the LH^{Penk}-LPAG circuit primarily governs HFD overconsumption following PSS exposure. Taken together, these data suggest that endogenous enkephalinergic systems in the LH circuitry have a major contribution to modulating post-threat effects on emotional overeating.

We presented the updated/new data in **Fig 5a-d, 5i-k, Fig. 6d-j in the revised manuscript**. We hope that we have now addressed the Reviewer's concern.

6. Chemogenetic activation of LH^{Penk} neurons acutely enhances HFD consumption. It would be interesting to explore

whether optogenetic activation of these neurons also induces HFD consumption (e.g. Supplemental Figure 5). Additionally, it is important to consider the differences between the acute modulation of LH^{Penk} neurons inducing HFD consumption and the 24-hour delay required for CORT (stress) to have a similar effect. Does the activation of LH^{Penk} neurons drive overconsumption of NC?

As requested, we employed an optogenetic approach to further confirm LH^{Penk} neuronal activation-driven HFD overconsumption. Using the Penk-Cre mice, we bilaterally expressed AAV-DIO-ChR2-eYFP into the LH followed by optic fiber implantation in the same site. We found that photoactivation of LH^{Penk} neurons at 20 Hz selectively increased the consumption for HFD, but not NC (**Extended Data Fig. 6g, h in the revised manuscript**), which is consistent with our previous data using the chemogenetic activation (**Fig. 4a-c in the revised manuscript**).

Regarding the Reviewer's point on *considering the differences between the acute modulation of LH^{Penk} neurons inducing HFD consumption and the 24-hour delay required for CORT (stress) to have a similar effect*, we were not able to observe the particular differences in behavioral outcomes on the surface. In terms of degree for HFD overconsumption or food preferences level, mice in both groups (acute modulation of LH^{Penk} neurons vs 24-h delayed after PSS/CORT) showed similar levels of overconsumption, which appears to be selective to HFD ($1.082 \pm 0.113\text{g}$ vs $1.095 \pm 0.0624\text{g}$, respectively), but not to NC (**Fig. 4a-c and Fig. 1g in the revised manuscript**).

It is possible that there are differences in engaging circuit components in each condition. Although LH^{Penk} neurons are the key player triggering emotional overconsumption, PSS/CORT-induced HFD overconsumption may include several circuit components, due to the general distribution of GR beyond the LH circuitry. However, the LH^{Penk} neuronal activation-driven HFD overconsumption is likely to be specified to the effects of LH^{Penk} neuronal circuitry. For example, in our data showing the behavioral responses to CTAP local infusion into the LPAG, we found that pharmacological inhibition of MORs in the LPAG strongly blocked the HFD overconsumption triggered by artificial activation of LH^{Penk} neurons (**Fig. 6f in revised manuscript**), whereas a relatively moderate reduction was observed in the PSS context (**Fig. 6j in revised manuscript**). Future work will need to parse the contributions of the holistic neural components in driving threat-induced palatable food overconsumption.

7. Kir2.1-mediated inhibition of LH^{Penk} neurons alleviated emotional overconsumption associated with negative emotional states in PSS mice. It would be interesting to investigate whether chronic manipulation of these neurons (in this case, inhibition of LHA^{Penk} neurons) impacts the animals' body weight or metabolism.

As we responded to comment #2, the effects of PSS on HFD overconsumption are likely to be most impactful within a certain time window like 24 h, instead of showing long lasting impacts during a chronic period. Investigating the effects of chronic manipulation of LH^{Penk} neuron on body weight or metabolism would be interesting, though we believe it is somewhat out of scope of this manuscript.

Reviewer #3 (Remarks to the Author):

In this paper they identified a population of proenkephalin-expressing LH neurons that increase engagement during high-fat diet eating 24 hours after a stress response-inducing exposure to predator odor and showed that chemogenetic activation of LHPenk neurons is sufficient to drive HFD overconsumption in an unstressed animal while also promoting a negative valence state as evidenced by formation of a conditioned place aversion. These findings suggest that stress-induced modulation of LHPenk activity via a GR-dependent mechanism induces negative affect to promote overconsumption of highly palatable food. Overall, the paper makes a compelling case for the GR-dependent mechanisms, but the lack of consideration for a role of Penk failed to directly link the genetic ID of the cells with anything meaningful, and simply treating the Enk as a marker is insufficient to support publication at this time. More evidence for specific role of the Enk would substantially improve the manuscript.

We thank the Reviewer for their time invested in making suggestions for improvements. We address each point below.

Major Considerations:

1. The role of enkephalin as a marker for this LH neuron population remains unclear. There is nothing which suggest that all LH neurons are not involved in the behavior; they show that the Penk LH population has little overlap with other prominently studied LH populations (Pmch, Hcrt, Lepr), but there aren't any control experiments showing that these other populations don't have the same increase in cFos during HFD after stress/activity during HFD first bout, etc. I think the easiest way to start getting at this would be in Figure 6j-k, what proportion of cFos+ neurons are also GFP-? We only get a quantification of Penk cells that are cFos+, it would be more convincing to show that non-Penk neurons in the LH are not activated by this condition. Otherwise, it feels like we are arbitrarily winnowing in on a 'subpopulation' without evidence that any of these outcomes are unique. Since there is also no data showing a functional role for enkephalin in this context (see point 2), I think there needs to be more evidence for why we think Penk is the marker here.

This point is well taken. As we responded to the Reviewer 1's comment # 3, we have compared *c-fos* induction levels between Penk LH vs. non-Penk LH population after PSS+HFD exposure. Given our data indicating that Lepr-expressing LH (LH^{Lepr}) neurons are mostly distinct from LH^{Penk} neurons (**Fig. 2g, h, and Extended Data Fig. 2a-c in the revised manuscript**), we examined whether the LH^{Penk} neurons constitute a special population engaged in PSS-induced HFD overconsumption compared to LH^{Lepr} neurons. We injected AAV expressing a Cre-dependent eGFP (AAV-DIO-eGFP) into the LH of Penk-Cre or Lepr-Cre mice to label the Penk- or Lepr-positive LH population, respectively. We then counted LH cells showing both *c-fos* immunoreactivity and eGFP fluorescence at 24 h post-PSS exposure. We observed that 29.91% of LH^{Penk} cells contained detectable *c-fos*+ nuclei in response to HFD eating, but to a lesser extent, only 10.72% of the LH^{Lepr} cells expressed *c-fos* under the same condition (**Fig. 5f, g in the revised manuscript**). After PSS+NC exposure, we were not able to find any changes in *c-fos* induction in both cell-type populations, suggesting that LH^{Penk} neurons play a more important role in triggering PSS-induced HFD overconsumption than LH^{Lepr} population.

Regarding Pmch- or Hcrt-expressing LH population, we were not able to apply the same strategy as described above, because we currently do not have validated Pmch-Cre or Hcrt-Cre mice. Moreover, it was difficult to apply double labeling immunofluorescence using anti-*c fos* (Cell Signaling Technology; Cat# 2250S) and well-characterized antibodies for Mch/Hcrt (Phoenix Pharmaceuticals INC; Cat# H-070-47 and Cat# H-003-30, respectively), because they all are raised in the same host species (rabbit). Therefore, we decided to perform additional experiments to substantiate the functional role for enkephalin in the PSS context (please see our responses to comment #2 below).

2. Any evidence of a functional role for LH enkephalin in stress-related binge-eating should be provided in at least one or two ways (to be rigorous). Since Penk is used as the marker identifying this population, this paper would be substantially more impactful and stronger if they can demonstrate that enkephalin is involved in these behaviors. If it is not, following experimental manipulations, then the authors will need to demonstrate why this particular population is special as compared to other LH populations (see above pt 1).

Some experimental ideas which would facilitate the above:

a. Conditional knockout of Penk in the LH, this gives you multiple options for evaluating the functional role of enkephalin

in this context. Test if PSS still produces higher HFD consumption and CPA. Test if PSS still increases cFos expression in the hypothalamus. Test if LH^{Penk} calcium activity is still elevated during the first HFD eating bout after stress. These will start answering if enkephalin has a role in the process, and if so at what point in the process. Is enkephalin involved in the sensitization of these neurons to stress during the actual stress exposure? Is it released during the consumption period post stress? Is it relevant in HFD and NC consumption or only HFD? Feasibility of CRISPR guide to induce Cre-dependent Penk cKO demonstrated in Castro Nature 2021.

b. Use the already obtained input/output tracing data to identify the pre/ post-synaptic opioid receptor population that these LH neurons would release enkephalin onto. This would open up many doors to test a functional enkephalin role (e.g. local infusion of MOPR agonist/antagonist in sufficiency and necessity of driving the PSS valence and consumption responses, selective knockout of receptor in the downstream population, record calcium activity of the downstream MOPR population during HFD eating post-stress to see if there are changes in activity corresponding with the increases in LH^{Penk} activity during the first feeding bout). I think this would also be the strongest narrative-wise, to be able to isolate endogenous opioid function and to also map a more complete circuit driving the behavior.

c. The simplest starting experiment: does treatment with an opioid receptor antagonist either before stress or before post-stress feeding prevent any of the phenotypes? This could at least be a starting point for identifying any opioid function in stress binge eating before spending too much time with the more careful experiments.

As suggested, we have performed several new experiments investigating the functional roles of Penk in the LH.

First, to test the necessity of LH endogenous enkephalin in potentiating HFD consumption after PSS exposure, we applied a CRISPR/SaCas9 viral-based system (AAV-CMV-FLEX-SaCas9-U6-sgPenk³; Addgene #159916) and disrupted enkephalin neuropeptide in the LH of Penk-Cre mice. Penk mRNA levels were measured using qRT-PCR and found to be reduced over 50-70% (**Fig. 5i, j in the revised manuscript**). At 24 h post-PSS exposure, we found that CRISPR-mediated reduction of Penk in the LH significantly alleviated PSS-induced HFD overconsumption as well as aversive emotional responses toward PSS-paired chamber in the CPA paradigm (**Fig. 5k-n in the revised manuscript**). These data suggested that LH endogenous enkephalin is necessary for mediating PSS-induced HFD overconsumption associated with a negative emotional state.

Second, using AAV shRNA-mediated Penk knockdown, we further confirmed that downregulation of enkephalin in the LH plays a significant role in preventing PSS-induced HFD overconsumption. We bilaterally injected AAV carrying a short hairpin RNA (shRNA) against Penk in a Cre-dependent manner (AAV-EF1a-DIO-EmGFP-Penk shRNA) into the LH of Penk-Cre mice and confirmed a remarkable reduction in Penk mRNA expression without accompanying changes in the Hcrtr, Pmch and Lepr in the virus injection area (**Fig. 5a, b in the revised manuscript**). Upon HFD exposure, we found that PSS mice expressing Penk shRNA showed attenuated HFD intake (**Fig. 5c, d in the revised manuscript**), consistent with our data using a CRISPR/SaCas9 viral-based system.

Third, given previous studies demonstrating that the enkephalin can act as a direct or indirect modulator for transmission of multiple neurotransmitter systems¹⁰, we reasoned that the LH enkephalin can contribute to change LH^{Penk} neuronal reactivity to PSS+HFD condition. We indeed found that the shRNA-mediated knockdown of Penk in the LH decreased *c-fos* induction in the LH^{Penk} neuronal population after the PSS+HFD exposure, whereas Penk-Cre mice injected with a control virus (AAV-DIO-eGFP) stably increased *c-fos* induction under the same condition (**Fig. 5f-h in the revised manuscript**). Together, these data confirmed that LH endogenous enkephalin is critical to potentiate LH^{Penk} neuronal reactivity to HFD, which is a hallmark of PSS mice exhibiting emotional overconsumption.

Fourth, the LPAG, one of the downstream structures of LH^{Penk} neurons (**Fig. 6a-c in the revised manuscript**), highly expresses μ -opioid peptide receptors (MORs) that are known to modulate binge eating behaviors^{4,5}. Because the enkephalin can act as a potent, efficacious endogenous agonist to MORs systems, we hypothesized that activation of MOR signaling in the LPAG is a critical component that relays the enkephalin information from LH^{Penk} neurons and leads to HFD overconsumption. Indeed, we found that HFD overconsumption driven by chemogenetic activation of LH^{Penk} neurons is strongly suppressed by intracerebral microinjections of the MOR-selective antagonist CTAP into the LPAG in a dose-dependent manner (**Fig. 6d-f in the revised manuscript**). Moreover, at 24 h post-PSS exposure, we found that a local infusion of the CTAP (1 μ g/side) into the LPAG normalized HFD consumption but not avoidance behaviors in the

CPA experiments (**Fig. 6i-n in the revised manuscript**), suggesting that MOR signaling in the LH^{Penk}-LPAG circuit primarily governs HFD overconsumption following PSS exposure, while alternative downstream targets of LH^{Penk} neurons may exist to drive PSS-induced negative emotional state.

Overall, these data indicate the functional relevance of the endogenous enkephalinergic systems in the LH circuitry in modulating post-threat effects on emotional overeating, supporting the ideas that LH^{Penk} neurons are indeed unique in mediating PSS-induced maladaptive behavioral changes.

We presented the updated and new data in **Fig. 5, 6 of the revised manuscript**. We hope that we have now addressed the Reviewer's concern.

3. The mifepristone microinfusion experiment in Figure 6 is somewhat unsatisfying in the sense that the antagonist will block all GRs in the LH and not just LHPenk GRs. How large is the GR+ Penk- population (this should be quantifiable from the dataset in 5c)? If small, then this is maybe less of a concern. If there are a significant number of non-Penk LHGR neurons however, then I think a follow-up experiment completing a conditional knockout of GR on LHPenk neurons is warranted.

As we responded to the Reviewer 1's comments #6 and 7, we reanalyzed Penk/GR FISH data to detect the extent of the overlap within the whole LH. We found that the majority of Penk+ neurons (79.48%) also expressed GR, while 19.36% of GR+ neurons co-expressed Penk. Because the GR+ Penk- cells seem to comprise a major population (80.64%), we decided to perform a follow-up experiment to address the role of GR *specifically expressed* in LH^{Penk} neurons.

As the Reviewer suggested, we have designed and tested several AAV constructs expressing shRNA against GR, but none of those constructs showed relevant knockdown efficiency. Instead of applying knockout/down of GR, we thus planned to *overexpress* GR *specifically* in LH^{Penk} cells and tested whether the overexpression of GR in LH^{Penk} cells can abolish GR antagonist (mifepristone)-induced behavioral normalization. To test this, a mouse cDNA for GR (also known as Nr3c1; NM 008173.4) was amplified by PCR and inserted into AAV vector to produce AAV-hsyn-DIO-GR. To validate specific overexpression, we stereotaxically injected AAV-hsyn-DIO-GR into the LH of Penk-Cre mice and confirmed that GR mRNA levels were robustly increased using both FISH experiments and qRT-PCR assay (**Fig. 7j and Extended Data Fig. 7f in the revised manuscript**). We indeed found that LH^{Penk} neuron-specific overexpression of GR prevented microinfusion of mifepristone from inhibiting the CORT-induced HFD overconsumption (**Fig. 7k in the revised manuscript**). This confirms that the effects of mifepristone on normalizing HFD intake are primarily mediated by inhibiting GR in the LH^{Penk} cells.

We presented the new data in **Fig. 7i-k and Extended Data Fig. 7f of the revised manuscript**. We hope that we have now addressed the Reviewer's concern.

4. In Figure 3, what is the explanation for using a novel object as the control for HFD instead of the NC food, which is used throughout the rest of the paper?

It is true that we used standard NC as a control in the food consumption tests, but NC alone was not sufficient to trigger some extent of interest or motion reactions in mice as they were always exposed to NC *ad libitum*. However, a novel object can promote active investigation including sniffing, approaching and exploring due to unfamiliarity. Given that HFD can be considered as a novel stimulus and that *in vivo* imaging experiments in **Figure 3** aimed to examine whether LH^{Penk} neuronal activity is correlated with sensitized responses to *HFD eating* after PSS exposure, we believed that a novel object would be better control than the NC, in order to rule out the effect of other accompanying behavioral reactions (e.g., sniffing, approaching and exploring with physical contacts) that also occurred when mice encountered HFD.

5. The effect in Figure 5j of a CORT-induced increase in LHPenk activity during 1st bout HFD feeding appears to be predominantly driven by 3 cells – it is unclear if this is spread across all 4 animals that were recorded or if these cells are from one animal. I think it would be more convincing to add a few more animals to these groups.

As requested, we performed additional experiments with one vehicle-treated and one CORT-treated Penk-Cre mice, thereby increasing the total cell numbers up to 51 and 68 cells, respectively. In this updated data set (**Fig. 7n in the**

revised manuscript), we found a stronger statistical significance [*** $p < 0.001$ for CORT-treated mice before vs. after 1st HFD eating bout; † $p = 0.021$ for veh- vs. CORT-treated mice after 1st HFD eating bout], which is strengthening our previous findings. Please note that CORT-treated mice consistently showed higher LH^{Penk} neuronal activity in response to HFD than vehicle-treated mice, although the degree of the enhancement is variable across the individual cells. In this additional experiment, we were able to add new data points showing the similar level of high responses as the 3 cells displayed in our original manuscript. We presented the updated data in **Fig. 7n in the revised manuscript**.

6. I'm unsure about using conditioned place aversion as a metric for persistent 'negative emotional valence.' When thinking about this in a real-world context, a sustained negative emotional state following a stressful event does not require re-exposure to the context where the stressful event occurred. To me the CPA is indicative of learning and memory as the mice formed an association between the context and the stimulus, as opposed to the authors' interpretation of CPA as a retention of negative emotions. Would those negative emotions exist if they weren't exposed to the stressful context again? The authors might consider other models of negative affect to bolster their results and conclusions.

We respectfully disagree on this point. In a real-world context, a sustained negative emotional state of psychiatric disease patients is often expressed by 'triggers' that can remind them of the stressful event they experienced. For example, one common symptom of posttraumatic stress disorder (PTSD) is flashbacks in which the patients express intense emotional reactions, such as fear, anxiety or anger, because they re-experience a past traumatic incident as if it is taking place in the current moment. Clinically, it has been reported that the PTSD flashbacks are often triggered by certain stimuli like particular places, people, or situations related to the trauma¹¹.

In the CPA paradigm, we found that mice showed active avoidance for the particular compartment paired with PSS. This paradigm is based on memory, but it is *more than* memory in that mice showed intense emotional reactions as if they encounter PSS again. To bolster our ideas, we have performed additional novel object recognition tests and examined cognitive ability and memory of mice. Especially in **Fig. 5m, n in revised manuscript**, we claimed that CRISPR-mediated reduction of Penk in the LH alleviated PSS-induced negative emotional state in the CPA paradigm, because mice given CRISPR infection no longer displayed avoidance behaviors toward PSS-paired chamber. We then questioned that *Could it be that they did not show active avoidance behavior because they were not remembering PSS?*

Using the same mice in **Fig. 5m, n in revised manuscript**, we thus performed novel object recognition tests to determine whether the reduction of avoidance behaviors is due to their memory loss. For novel object recognition tests, two identical objects were placed in each corner of the arena, and mice were allowed to explore the area for 10 min (training session). Twenty-four hours after the training, mice were placed in the arena where one of the two objects was replaced with a novel object having different color and shape. Discrimination rate was calculated as [(time spent in a novel object area) / (time spent in a novel object area + time spent in a familiar object area) x 100 (%)]. We found that the mice with CRISPR infection showed a normal ability to recognize a novel object (**please see a figure below**; two-tailed unpaired t -test, $t_{10} = 0.820$, $p = 0.431$), indicating that they could show attenuated avoidance behaviors in CPA paradigm, though they have normal functions in learning and memory. Together, we speculate that PSS and LH^{Penk} neuronal manipulation can regulate *negative emotional component*, rather than the *memory component*, which are separable, but existing together in the CPA paradigm.

7. What was the justification for using Kir2.1 as a chronic silencing method in Figures 4h-j? It would seem more compelling to use a more temporally-precise silencing tool (e.g., halorhodopsin) to identify the relevant times where LHPenk activity is necessary to induce HFD overconsumption.

As we responded to the Reviewer 1's comment #5, PSS-induced HFD overconsumption requires time to take hold as evidenced in our data showing that mice exhibited HFD overconsumption at 24 h after, but not immediately after the PSS exposure (**Fig. 1f, g in the revised manuscript**). Moreover, CORT-driven HFD overconsumption is clearer a day after, but not immediately after the CORT administration (**Extended Data Fig. 7b in the revised manuscript**), suggesting that time delay between the PSS-induced CORT elevation and subsequent HFD exposure would allow for mice to develop and display their heightened craving for HFD eating. We speculated that LH^{Penk} neurons may be a key component that has undergone CORT-induced neuronal adaptation during the delay between the PSS exposure and the measurement of HFD intake. It is also likely that any active adaptive processes activating LH^{Penk} neurons occur during this interval.

We performed chronic silencing experiments with Kir2.1 instead of temporally-precise silencing tool, in order to block the potential LH^{Penk} neuronal activity changes during the interval (24 h) without specific information on *when* the maladaptive responses of PSS-primed LH^{Penk} neurons will be expressed. Afterwards, through real-time monitoring of *in vivo* LH^{Penk} neuronal activity, we observed that the increased activity of PSS-primed LH^{Penk} neuronal activity is prominent upon HFD exposure. Although identifying the relevant times when LH^{Penk} activity is necessary to induce HFD overconsumption is an interesting topic, we believe that this is beyond the scope of this manuscript.

8. There needs to be a greater distinction between what the authors refer to as “negative emotion” and fear memory. How can they be sure that their results are due to LHPenk involvement in negative emotions and not some other aspect of aversion? This is a related point to the one above about negative affect/valence.

Please note that we substantiated our idea by presenting ‘real-time’ place test (RTPT) data in our original manuscript (**Extended Data Fig. 6c, d in revised manuscript**). Using Penk-Cre mice, we expressed the Cre-dependent Channelrhodopsin2 (ChR2) in the LH followed by the implantation of an optic cannula in the same site. We found that the ChR2-expressing Penk-Cre mice avoided the side of the RTPT apparatus paired with photostimulation (10-ms pulses of 473-nm light at 20 Hz), but control mice did not, consistent with our data obtained using chemogenetic activation (**Fig. 4d, e in revised manuscript**). It is important to note that, in this RTPT paradigm, the mice ‘instantly’ showed avoidance for the side paired with *real-time* optogenetic stimulation, indicating that LH^{Penk} neurons represent a negative emotional valence, rather than a fear memory that is consolidated over time.

9. What is the point of the tracing/circuit data? This data feels out of place in the paper and largely ignored throughout the rest of the experiments. The results from the *in situ* hybridization experiments are also largely ignored in the remainder of the text. How these results help the authors gain insight into the role of LHPenk neurons remains unclear. Rearranging the order in which the data is presented might help address these issues.

In this revised manuscript, we have added new data sets indicating the functional roles of LH^{Penk}-LPAG circuitry in modulating PSS-induced HFD overconsumption (**Fig. 6d-j in revised manuscript**). Accordingly, we relocated the tracing/circuit data of LH^{Penk} neuronal efferent projections from Extended Data Fig. 2a-c in original manuscript to **Fig. 6a-c in revised manuscript**.

10. Does the GR antagonist block the effects of corticosterone on LHPenk activity following PSS and during GFD consumption? The only experiment done using a combination of the GR antagonist and corticosterone focused on their effects on consumption, not neuronal activity.

In response to the Reviewers' question, we have performed additional experiments. After injecting AAV-DIO-eGFP into the LH of Penk-Cre mice, we implanted a cannula into the same site for the local infusion of mifepristone (**Extended Data Fig. 8d in the revised manuscript**). We then carried out *c-fos* immunostaining and quantified *c-fos*-positive cells among eGFP-labelled LH^{Penk} neurons after the drug administrations. We found that up to 28.63% of LH^{Penk} neurons

showed HFD-induced *c-fos* expression 24 h after veh (μ -infusion) + CORT (s.c.) administration, but a local infusion of mifepristone blocked the induction of *c-fos*, resulting in only 5.48% of LH^{Penk} neurons showing *c-fos* expression under the same condition (**Extended Data Fig. 8e, f in the revised manuscript**). These data support the hypothesis that CORT-induced activation of GR signaling in the LH is required for sensitizing LH^{Penk} neuronal activity to HFD, leading to palatable food overconsumption. We have added the data in **Extended Data Fig. 8d-f in the revised manuscript**.

Minor Considerations:

-The introduction is missing a discussion of any previous investigations of LH^{Penk} existence/function, and only talks about enkephalin's role in other brain regions. What evidence is there that Penk expresses in the LH, what is known already about these neurons?

At this moment (July/14/2023), a Pubmed search using the terms "proenkephalin and lateral hypothalamus" yields only 4 items. One recent study² delineated the roles of Penk-expressing ventrolateral PAG neurons receiving inputs from the LH; the other three of early studies¹²⁻¹⁴ investigated Penk distribution/mRNA levels in the rat brain during developmental stages or hypertension, yet the precise roles of Penk in the LH had not been clarified. It is surprising that little is known about the functional relevance of the LH^{Penk} neurons, although enkephalin peptides in other brain regions are heavily studied and known to be involved in drug addiction^{15,16} and affective disorders^{17,18}. To the best of our knowledge, our results in this manuscript are the first evidence demonstrating the functional role of LH^{Penk} neuronal population in eliciting threat-induced overeating associated with negative emotional valence.

Regarding the previous investigations of Penk mRNA *existence* in the LH, we have added the citations of early studies as mentioned above in our **Introduction section of revised manuscript (page 4)**.

-The LH is a neuroanatomical hub responsible for regulating diverse primitive behavioral states, such as feeding, as well as approach and avoidance behaviors." This claim on page 8 is missing several citations. Primarily Nieh et al. *Neuron*, 2016, and Marino et al., *Proc Natl Acad Sci USA*, 2020, which show LH circuits involved in food approach, negative affect, approach/avoidance behavior.

We appreciate the Reviewer's point. In our revised paper, we added those citations in **page 8**.

-Switch the color scheme on the pie charts in Figure 2 so that the +/+ group is always in color, and the +/- group is always in grayscale. It's confusing to swap what corresponds with colored vs grayscale between 2h and 2j, and I think the default when glancing at a figure is to assume the group in color is the positive condition (in this case, both markers co-localize) and the grayscale condition is the 'null' (Penk does not co-localize with the other marker). I think that will highlight the idea that the Penk neurons are mainly a standalone population, regardless I think it should be consistent throughout this figure and not switch once you get to Vgat/Vglut2.

We have corrected this and updated the color compositions of **Fig. 2 in revised manuscript**.

-"Consumption of palatable foods can activate neural activity of the hypothalamus" (Page 7). Is this claim meant in a no-stress context? If so, the results in Extended Data Figure 1 do not agree with this statement, where HFD did not increase neural activity in LH, PVN, DMH, VMH, or ARC relative to NC diet in the unstressed mice.

We have clarified and modified the claim as follows: "*Exposure to stressful situations can promote palatable food consumption via increasing the hypothalamic circuit activity involved in controlling eating behaviors*" (**page 7 of the revised manuscript**).

-For the anterograde and retrograde tracing experiments in Extended Data Figure 2, can the authors show the negative data that the DIO viral injections were contained specifically to the LH? The data in Figure 2a suggest that there are other Penk-expressing hypothalamic subregions, so the tracing could potentially pick up inputs/outputs not specific to the LH if the viral expression was not restricted.

The injection sites were confirmed in all animals by preparing coronal sections that contains the LH, and animals with incorrect injection placement were excluded from the analyses. After each experiment, the extent of viral transduction spread was examined at the conclusion and viral transduction was validated whether it was limited to the LH without off-target effects in other adjacent areas, such as the Arc, VMH and DMH. The inclusion criteria were applied when the virally labeled neurons were located in the LH reference area; that is, lateral and up to 0.1 mm medial to the fornix or up to 0.2 mm above or below the fornix. If the viral expression was found outside this reference area or the viral transduction was weak in the LH (covering less than 50% of the total LH area), the mice were excluded from the final dataset, which was determined by two experimenters who were blinded to the experimental design. Please note that, in our original manuscript, we presented whole hypothalamus images indicating the restricted DIO viral expression within the LH (Fig 3b, Fig 4a, h, Fig. 6b, Fig 8i; Extended Data Fig. 3c, 6c in the revised manuscript). We also presented additional supporting data as below; A left one indicates the whole hypothalamus image of Penk-Cre mice given AAV-DIO-eGFP injection into the LH; A right one is negative data showing the brain slice of C57BL/6J mice that we injected the same DIO virus into the LH, confirming the specificity of our DIO viral system (please see figures below; Scale bars, 200 μm).

-Can the author's show individual data for Figures 4e-g? The medians appear identical for the hM3D group pre/post-test.

As we responded to the Reviewer 1's comment # 9, we added more n numbers up to 9 mice per group and reanalyzed the data with Two-way RM ANOVA ($F_{(1,16)} = 5.011$, $p = 0.040$) followed by Bonferroni post hoc test to compare the results in pre vs. post x AAV-DIO-mCherry vs. AAV-DIO-hM3D, strengthening our previous conclusion. We also presented a graph showing individual data points as below.

-In Figures 5f-g, using L and H to denote the low and high doses of mifepristone is overly confusing, especially since this notation isn't used in the main results text. Just say 1 and 10 ng on the figure.

We appreciate the Reviewer's point. In **Fig. 7g, h and 8c-e of our revised paper**, we clarified the drug dose with 1 ng or 10 ng, instead of denoting them with L and H.

-Does mifepristone treatment prior to PSS have an effect on HFD consumption immediately following PSS? Since PSS only had an effect on delayed HFD and not immediate HFD (Figure 1f), it would be more convincing that GR is the responsible mechanism for this effect if the authors can show that mifepristone also does not impact HFD if assessed immediately after PSS.

In response to the Reviewer's question, we have performed additional experiments to examine whether the mifepristone treatment into the LH alters HFD intake immediately after PSS. 30 min prior to PSS exposure, we performed an intracranial microinfusion of either vehicle or mifepristone (1 ng/side) and subjected those mice to both HFD and NC *immediately* after PSS exposure. We were not able to observe any differences in food intake levels regardless of mifepristone treatment (**please see figures below**), supporting the idea that GR signaling in the LH is the responsible mechanism for delayed onset of HFD overeating after the PSS exposure.

-In Extended Data Figures 6m-n, the authors should add a CORT NC control group to show that CORT doesn't just generally increase LHPenk activity, and that the effect of the CORT HFD group is actually specific to the HFD.

Please note that, in our original manuscript, we presented data showing that CORT pretreatment itself did not affect baseline *in vivo* activity of LH^{Penk} neurons under the NC condition (**Extended Data Fig. 7i, j in the revised manuscript**). In addition, we have further confirmed this by performing additional *c-fos* immunostaining, when we responded to the Reviewer's comment #10. We have added this new data in the **Extended Data Fig. 8d-f in the revised manuscript**.

-Page 10, Line 1 : Replace "This led us..." with "These results led us..."

We have replaced the phrase in **page 12 of the revised manuscript** as suggested.

-Figure 4i+j , 6c+d : Symbols depicting the data points are empty instead of filled in like on all other figures. Choose between filled in or empty for all graphs.

We have changed individual data points to be filled in **Fig. 4i, j and 8c, d in the revised manuscript**. We also displayed more consistency in all graphs of the revised manuscript.

-Extended Data Fig. 5g : Squares are empty instead of filled in compared to 5h depicting the same data.

We have corrected this and displayed consistency between **Extended Data Fig 6j and k in the revised manuscript.**

References

1. Bonnavion, P., Jackson, A. C., Carter, M. E. & De Lecea, L. Antagonistic interplay between hypocretin and leptin in the lateral hypothalamus regulates stress responses. *Nat. Commun.* **6**, (2015).
2. Shin, S. *et al.* Early adversity promotes binge-like eating habits by remodeling a leptin-responsive lateral hypothalamus–brainstem pathway. *Nat. Neurosci.* **2022** *261* **26**, 79–91 (2022).
3. Castro, D. C. *et al.* An endogenous opioid circuit determines state-dependent reward consumption. *Nat.* **2021** *5987882* **598**, 646–651 (2021).
4. Nathan, P. J. & Bullmore, E. T. From taste hedonics to motivational drive: central μ -opioid receptors and binge-eating behaviour. *Int. J. Neuropsychopharmacol.* **12**, 995–1008 (2009).
5. Gutstein, H. B., Mansour, A., Watson, S. J., Akil, H. & Fields, H. L. Mu and kappa opioid receptors in periaqueductal gray and rostral ventromedial medulla. *Neuroreport* **9**, 1777–1781 (1998).
6. Burger, K. S., Sanders, A. J. & Gilbert, J. R. Hedonic Hunger Is Related to Increased Neural and Perceptual Responses to Cues of Palatable Food and Motivation to Consume: Evidence from 3 Independent Investigations. *J. Nutr.* **146**, 1807–1812 (2016).
7. Dallman, M. F. Stress-induced obesity and the emotional nervous system. *Trends in Endocrinology and Metabolism* (2010) doi:10.1016/j.tem.2009.10.004.
8. Panettieri, R. A. *et al.* Non-genomic Effects of Glucocorticoids: An Updated View. *Trends Pharmacol. Sci.* **40**, 38–49 (2019).
9. Krugers, H. J. & Hoogenraad, C. C. Hormonal Regulation of AMPA Receptor Trafficking and Memory Formation. *Front. Synaptic Neurosci.* **1**, (2009).
10. Rysztak, L. G. & Jutkiewicz, E. M. The role of enkephalinergic systems in substance use disorders. *Front. Syst. Neurosci.* **16**, (2022).
11. Kolk, B. van der. Posttraumatic stress disorder and the nature of trauma. *Dialogues Clin. Neurosci.* **2**, 7 (2000).
12. Boone, J. B. & McMillen, D. Proenkephalin gene expression is altered in the brain of spontaneously hypertensive rats during the development of hypertension. *Brain Res. Mol. Brain Res.* **24**, 320–326 (1994).
13. Laurent-Huck, F. M., Stoeckel, M. E. & Felix, J. M. Ontogeny of proenkephalin gene expression in the rat hypothalamus. *Brain Res. Dev. Brain Res.* **62**, 33–43 (1991).
14. Merchenthaler, I., Maderdrut, J. L., Altschuler, R. A. & Petrusz, P. Immunocytochemical localization of proenkephalin-derived peptides in the central nervous system of the rat. *Neuroscience* **17**, (1986).
15. Mongi-Bragato, B., Avalos, M. P., Guzmán, A. S., Bollati, F. A. & Cancela, L. M. Enkephalin as a Pivotal Player in Neuroadaptations Related to Psychostimulant Addiction. *Front. Psychiatry* **9**, 222 (2018).
16. Rysztak, L. G. & Jutkiewicz, E. M. The role of enkephalinergic systems in substance use disorders. *Front. Syst. Neurosci.* **16**, (2022).
17. Nam, H. *et al.* Reduced nucleus accumbens enkephalins underlie vulnerability to social defeat stress. *Neuropsychopharmacology* **44**, 1876 (2019).
18. Schwartz, J.-C., van Amsterdam, J., Llorens-Cortes, C. & Costentin, J. Endogenous Enkephalins, Depression and Antidepressants. *New Concepts Depress.* 247–259 (1988) doi:10.1007/978-1-349-09506-3_23.

REVIEWERS' COMMENTS

Reviewer #1 (Remarks to the Author):

The authors have done a thorough job in addressing the reviewer's comments. I now support its publication in Nature Communication.

Reviewer #2 (Remarks to the Author):

The authors did a great job of addressing my concerns over the last version. The data are beautiful, and the conclusions are justified. Congratulations to the authors for this exciting and compelling study.

Reviewer #3 (Remarks to the Author):

The authors have done a commendable job addressing each and every concern, and now the paper is even more exciting and thorough than before. The finding especially that Penk from the LH to PAG is mediating the behavioral effect via MOR interactions is very novel and important. I congratulate the authors on a very nice body of work!

REVIEWERS' COMMENTS

Reviewer #1 (Remarks to the Author):

The authors have done a thorough job in addressing the reviewer's comments. I now support its publication in Nature Communication.

Reviewer #2 (Remarks to the Author):

The authors did a great job of addressing my concerns over the last version. The data are beautiful, and the conclusions are justified. Congratulations to the authors for this exciting and compelling study.

Reviewer #3 (Remarks to the Author):

The authors have done a commendable job addressing each and every concern, and now the paper is even more exciting and thorough than before. The finding especially that Penk from the LH to PAG is mediating the behavioral effect via MOR interactions is very novel and important. I congratulate the authors on a very nice body of work!

We thank all Reviewers for their supportive and positive comments about our manuscript (NCOMMS-23-12381A). We also appreciate all the Reviewer for time and effort they allocated to providing a thorough review, which helped to improve the paper.